# Divergent iron dissolution pathways controlled by sulfuric and nitric acids from the ground-level to the upper mixing layer

Guochen Wang[1], Xuedong Cui[2], Bingye Xu[3], Can Wu[4], Minkang Zhi[1], Keliang Li[1], Liang Xu[5], Qi Yuan[6], Yuntao Wang[7], Yele Sun[8], Zongbo Shi[9], Akinori Ito[10], Shixian Zhai[11], Weijun Li[1,8*]

[1]State Key Laboratory of Ocean Sensing and Department of Atmospheric Sciences, School of Earth Sciences, Zhejiang University, Hangzhou 310027, China
[2]Hangzhou Meteorological Bureau, Hangzhou 310051, China
[3]Ecological and Environmental Monitoring Center of Zhejiang Province, Hangzhou 310007, China
[4]Key Lab of Geographic Information Science of the Ministry of Education, School of Geographic Sciences, East China Normal University, Shanghai 210062, China
[5]College of Sciences, China Jiliang University, Hangzhou 310018, China
[6]College of Environmental Science and Engineering, Ocean University of China, Qingdao 266100, China
[7]State Key Laboratory of Satellite Ocean Environment Dynamics, Second Institute of Oceanography, Ministry of Natural Resources, Hangzhou 310012, China
[8]State Key Laboratory of Atmospheric Boundary Layer Physics and Atmospheric Chemistry, Institute of Atmospheric Physics, Chinese Academy of Sciences, Beijing 100029, China
[9]School of Geography, Earth and Environmental Sciences, University of Birmingham, Birmingham B17 8PS, UK
[10]Yokohama Institute for Earth Sciences, Japan Agency for Marine-Earth Science and Technology (JAMSTEC), Yokohama, Kanagawa 236-0001, Japan
[11]Earth and Environmental Sciences Programme and Graduation Division of Earth and Atmospheric Sciences, Faculty of Science, The Chinese University of Hong Kong, Sha Tin, Hong Kong SAR, China

*Correspondence to*: Weijun Li (liweijun@zju.edu.cn)

**Abstract.** Iron (Fe) plays a crucial role in the global biogeochemical cycle, marine ecosystems, and human health. Despite extensive research on Fe dissolution, the understanding of the mechanism of the Fe acidification process remains highly controversial. Here, we revealed significant differences in Fe acid dissolution between the upper mixing layer and the ground-level of a megacity. The results showed that air masses with elevated $n[SO_4^{2-}]/n[NO_3^-]$ ratios (5.4 ± 3.7) yielded more enhanced iron solubility (%$Fe_S$, 8.7 ± 2.4%) in the upper mixing layer after atmospheric aging compared to those (1.6 ± 0.7 and 3.3 ± 0.4%, respectively) at the ground-level near source regions of acidic gases. Further analysis suggested that Fe dissolution is primarily driven by sulfuric acid in the upper mixing layer different from nitric acid at the ground-level, attributing to the aging processes of acidic species during long-range transport. %$Fe_S$ also exhibits a clear size dependence: sulfuric-acid dominates in submicron aerosols ($D_p$ <1 μm), leading to elevated %$Fe_S$ (3.5 ± 3.9%), whereas alkaline mineral dust in supermicron particles ($D_p$ >1 μm) neutralizes nitric acid and suppresses Fe dissolution (1.8 ± 2.2%). This finding highlighted that sulfuric acid dominates Fe acidification process in the upper layer and submicron particles, but the contribution of nitric acid to Fe dissolution at the ground-level is equally important. Our study provides new data sets for testing atmospheric model's capability to simulate dissolved Fe concentration and deposition and will help to improve the accuracy of Fe solubility predictions.

# 1 Introduction

Iron (Fe) is an ubiquitous but essential element for life and plays a crucial role in the global biogeochemical cycle, marine ecosystems, and human health (Mahowald et al., 2009; Martin, 1990; Boyd and Ellwood, 2010). Despite its abundance in the earth's crust, Fe is often a growth limiting factor, controlling primary productivity in up to one-third of the world's oceans with high nutrient low chlorophyll (HNLC), thereby influencing carbon sequestration (Mahowald et al., 2009; Martin, 1990). Additionally, redox cycling of Fe affects the formation of reactive oxygen species (ROS) in aqueous reactions, causing adverse health effects (Vidrio et al., 2008; Chen et al., 2024a). The vital role of Fe in global climate change and human health underscores the need to understand the mobilization and dissolution of Fe during atmospheric transformations (Jickells et al., 2005; Martínez-García et al., 2014; Boyd et al., 2007; Fang et al., 2017).

The acidification process of insoluble Fe-containing aerosols by acids (e.g., $H_2SO_4$ and $HNO_3$) has been identified as the controlling factor in Fe solubility (%$Fe_S$) in particles (Baker et al., 2021; Ingall et al., 2018; Longo et al., 2016; Meskhidze, 2005; Shi et al., 2015). In this process, acids condense on the surface of insoluble Fe-containing particles and elevate the aerosol acidity. Both laboratory studies (Cwiertny et al., 2008; Shi et al., 2015; Ito and Shi, 2016) and field observations (Lei et al., 2023; Oakes et al., 2010) have shown that the heterogeneous reaction of $SO_2$ on the surface of mineral dust forms an extremely acidic environment that promotes Fe dissolution. Zhuang et al. (1992) initially proposed the hypothesis of the coupling and feedback mechanism between Fe and sulfur during the long-range transport of Asian dust, underscoring the crucial role of sulfuric acid in dissolving Fe. Similar findings have been documented during the transport of North Africa dust to the Atlantic Ocean (Zhu et al., 1997), in the North India Ocean (Bay of Bengal) (Srinivas et al., 2011), in major U.S. cities like Atlanta (Wong et al., 2020; Oakes et al., 2012), in offshore regions of China (such as the Yellow Sea) (Li et al., 2017; Meskhidze, 2003) and its cities (Zhu et al., 2020), and even within simulated cloud processes (Wang et al., 2019; Chen et al., 2012). However, other studies also indicated that nitric acid can equally or more effectively promote Fe dissolution than sulfuric acid (Zhu et al., 1997; Sakata et al., 2023; Rubasinghege et al., 2010). For instance, recent work by Zhu et al. (2020) determined that nitric acid elevates %$Fe_S$ in urban environments in eastern China, emphasizing its contribution to Fe acid dissolution.

Despite extensive previous research on Fe dissolution, understanding the role of proton-promoted process remains highly controversial. Atmospheric acidification accelerates Fe dissolution, primarily depending on the type and relative abundance of acids and their aging process. Variations in emissions of acidic gases such as $SO_2$ and $NO_x$ (=$NO+NO_2$) will consequently affect the formation of $H_2SO_4$ and $HNO_3$, creating an acidic environment (Rubasinghege et al., 2010; Ooki and Uematsu, 2005). Moreover, the aging process of acidic species modulates atmospheric chemistry (e.g., aerosol acidity) during long-range transport, thereby influencing Fe dissolution process (Baker et al., 2021; Li et al., 2017; Xu et al., 2023). To estimate proton levels, the atmospheric chemistry models consider thermodynamic processes involving the sulfuric acid and nitric acid systems (Ito, 2011; Myriokefalitakis et al., 2022). However, some global aerosol models (e.g., MATCH, MIMI and BAM-Fe) simplify the calculation of pH values for proton-promoted Fe dissolution by setting the pH solely from the sulfate-

to-calcite ratio (Hamilton et al., 2019; Scanza et al., 2018; Luo et al., 2005). With sustained sustainable reductions of $SO_2$ emissions, $NO_x$ has increasingly replaced $SO_2$ as the predominant inorganic acid source at the ground-level in China and other countries over the past two decades (Uno et al., 2020; Van Der A et al., 2017; Zheng et al., 2018; Geng et al., 2024). Such simplification cannot truly reflect proton levels and Fe dissolution rate under the regime shift from sulfuric acid to nitric acid (Ito and Xu, 2014). The accuracy of Fe simulation needs to be further verified. Furthermore, existing research

predominantly focuses on surface-level chemistry, neglecting the upper mixing layer where regionally aged aerosols reside. Because vertical transport significantly alters aerosol composition, the lack of altitude-resolved data limits the accuracy of atmospheric models. Investigating this vertical disparity is essential to constrain altitude-dependent mechanisms and improve model accuracy. Here, we raise the issue of whether the key chemical processes governing Fe dissolution differ between near-surface and the upper mixing layer in eastern China.

To answer this question, we designed field observations in the upper layer of a mountain and at the ground-level in a megacity to clarify how inorganic acids and their atmospheric aging influence Fe dissolution. By distinguishing air masses arriving at the two altitudes and examining their respective $n[SO_4^{2-}]/n[NO_3^-]$ ratios, we systematically compared the acid aging processes and associated Fe acidification under these contrasting atmospheric environments. Our results reveal distinct %$Fe_S$ levels between the upper layer and ground-level, driven by differing acid-processing pathways: sulfuric acid-

dominated Fe dissolution in the upper layer (due to longer atmospheric aging) versus nitric acid-dominated processing at the ground-level.

## 2 Data and methods

### 2.1 Sample collection

    The field campaigns were conducted during the summer of 2021 in a mountain site (Mt. Daming, 30.03°N, 119.00°E,

1483 m) and in the megacity of Hangzhou (30.30°N, 120.09°E, 6 m) (Fig. S1 in the Supplement). The mountain serves as a background environment and is sensitive to the transport of air pollutants from outside. Its high altitude makes it an ideal location for assessing the impacts of anthropogenic emissions in the upper layer. Since mixed-layer height (MLH) data were not available at Mt. Daming during the sampling period, we referenced MLH observations (CL51, Vaisala, Finland) from Hangzhou. The results show that the MLH in Hangzhou remained below ~1500 m for most of the time (~92%; Fig. S2).

Given the relatively short distance between the two sites (~110 km), the MLH in Hangzhou is considered a reasonable proxy for that at Mt. Daming. Therefore, owing to its elevation (~1500 m), the mountain site can be considered representative of the upper mixing layer. The urban site in Hangzhou, with a population of 12.52 million by the end of 2023 (Hangzhou Municipal Government, 2024), is one of the densely populated regions in the Yangtze River Delta (YRD).

    The field sampling was conducted from July 17 to August 19, 2021 for Mt. Daming and from September 11 to 21, 2021

for Hangzhou. A medium-flow total suspended particle (TSP) sampler (TH-16A, Wuhan Tianhong Instrument Co., Ltd, China) with a sampling flow rate of 100 L min$^{-1}$ was deployed to collect aerosol particles on quartz filters. Prior to sampling,

the filters were pre-combusted at 600 °C for 6 h to eliminate potential organic contamination. All samples were stored at −20 °C until further analysis. Ascribed to the rainy season (June−October), the sampling was temporarily stopped due to rain events. Totally, seven samples were obtained from mountain site and seven samples were collected from Hangzhou city.

Detailed sampling information is presented in Table S1 in the Supplement. A ten-stage cascade impactor (MOUDI 120R, MSP corporation, USA), operating at a flow rate of 30 L min$^{-1}$, was employed to collect size-resolved aerosol samples with cut points of 18, 10, 5.6, 3.2, 1.8, 1.0, 0.56, 0.32, 0.18, 0.10, and 0.056 μm. Three sets of samples were collected at Mt. Daming during the periods of July 29 to August 2, August 8−9, and August 18−19, 2021. Due to the maintenance of instrument, however, no MOUDI samples were collected in Hangzhou. All instruments were installed on open ground in

front of the monitoring site.

## 2.2 TSP and water-soluble inorganic ions measurements

TSP were determined gravimetrically using an ultra-high-resolution balance (Sartorius Lab Instruments GmbH & Co, Germany). Prior to sampling, blank quartz filters were conditioned for 24 h under controlled temperature and relative humidity (25 °C and 50% RH) and then weighed. After sampling, the filters were reconditioned under the same conditions

and reweighed. The difference between the post-sampling and pre-sampling filter masses was taken as the TSP mass. The water-soluble inorganic ions (WSIIs), including $SO_4^{2-}$, $NO_3^-$, $Cl^-$, $Na^+$, $NH_4^+$, $K^+$, $Mg^{2+}$ and $Ca^{2+}$ in TSP and size-resolved aerosols (MOUDI) were analyzed with an ion chromatography (DIONEX ICS-600). The detection limits of the measured $SO_4^{2-}$, $NO_3^-$, $Cl^-$, $Na^+$, $NH_4^+$, $K^+$, $Mg^{2+}$ and $Ca^{2+}$ are 0.021, 0.008, 0.010, 0.018, 0.006, 0.006, 0.009 and 0.022 μg mL$^{-1}$, respectively. More information about the laboratory chemical analyses can be found in our previous studies (Liu et al., 2022a;

Zhu et al., 2020). To distinguish the contribution of anthropogenic emissions to sulfate, sulfate associated with sea salt was subtracted from the measured sulfate in this study. The mass concentration of the non-sea salt sulfate (nss-$SO_4^{2-}$) can be estimated as follows:

$$nss\text{-}SO_4^{2-} = SO_4^{2-} - 0.25 \times Na^+ \qquad (1)$$

where $SO_4^{2-}$ and $Na^+$ are the mass concentrations of $SO_4^{2-}$ and $Na^+$, respectively. 0.25 is the mass ratio of $SO_4^{2-}$ to $Na^+$ in

pure seawater (Kunwar and Kawamura, 2014). Therefore, the sulfate used in the analysis is nss-$SO_4^{2-}$.

## 2.3 Measurements of total and dissolved Fe

The total iron (Fe$_T$) in TSP and size-resolved aerosols were measured non-destructively using an energy dispersive X-ray fluorescence (ED-XRF) spectrometer (Epsilon 4, PANalytical, Netherlands). A standard reference material (SRM 2786, National Institute of Standards and Technology (NIST), USA) was used to calibrate the instrument before sampling analysis.

The measured Fe concentrations were within the NIST certified values, with relative errors between measured and standard values below 10 %. Soluble iron (Fe$_S$) in the samples was measured using the ferrozine technique following the procedures of Zhu et al. (2022) and Zhi et al. (2025). Briefly, (1) two circular sections (radius = 8 mm) were cut and placed in

polypropylene bottles with 20 mL of ammonium acetate solution (0.5 mM, pH ~4.3); (2) after 60 min of sonication, the extracts were filtered through a 0.22 µm PTFE syringe filter (Tang et al., 2025); (3) the pH of the solution was adjusted to ~1.0 using 150 µL of concentrated HCl and stored at 4 °C before further analysis; (4) starting to measure the solution, a 0.01 M ascorbic acid was added to the solution and held for 30 minutes to ensure the complete reduction of Fe(III) to Fe(II); (5) adding 0.01M ferrozine solution; (6) adjusted the solution to ~pH 9.5 using ammonium acetate buffer. The absorbance of the solution was measured at 562 nm (max light absorption) and 700 nm (background) (Oakes et al., 2012) by using an UV-Visible spectrophotometer (UV-Vis, Specord 50 Plus, Analytik Jena Instruments, Germany). Ultra-grade ammonium Fe(II) sulfate hexahydrate (Sigma-Aldrich, St. Louis, USA) was used for Fe(II) standards. The concentration of Fe(II) obtained from the standard curve was the concentration of dissolved Fe.

The quality assurance/quality control (QA/QC) procedures for soluble Fe measurement included the following steps: (1) the spectrophotometer was powered on and stabilized for 6–8 hours prior to use; (2) the instrument was calibrated using seven ammonium Fe(II) sulfate standards with concentrations of 5, 10, 20, 40, 60, 80, and 100 ng mL$^{-1}$. The absorbance of each standard was measured at 562 nm ($I_{562}$) and 700 nm ($I_{700}$), and the absorbance difference ($\Delta I = I_{562} - I_{700}$) was correlated with Fe(II) concentration. When the coefficient of determination ($R^2$) exceeded 0.995, the instrument was considered properly calibrated; (3) prior to sample analysis, a 0.1 mol L$^{-1}$ HCl solution (pH = 1) was used as the reference; once the absorbance was displayed nearly to zero, the samples could be measured. (4) the average soluble Fe on blank filters was ~0.6 ng cm$^{-2}$, determined from two sets of TSP and one set of size-resolved samples. The average contribution of filter blanks accounted for less than 2% of the measured sample concentrations. All samples were corrected by subtracting the filter blank values. In this study, the calibration yielded an $R^2$ of 0.998 and the absorbance of reference solution displayed zero, indicating excellent instrumental stability and measurement reliability. Iron solubility was calculated using the following equation:

$$\%\mathrm{Fe_S}\ (\%) = \frac{\mathrm{Fe_S}}{\mathrm{Fe_T}} \times 100\% \quad (2)$$

where $\mathrm{Fe_S}$ and $\mathrm{Fe_T}$ are the soluble Fe and total Fe concentrations, respectively.

**2.4 Aerosol pH estimation**

The ISORROPIA II thermodynamic equilibrium model (https://www.epfl.ch/labs/lapi/models-and-software/isorropia/) was applied to simulate aerosol pH and liquid waters in TSP and size-resolved aerosols by using inorganic chemical species ($SO_4^{2-}$, $NO_3^-$, $NH_4^+$, $Cl^-$, $Na^+$, $K^+$, $Mg^{2+}$, $Ca^{2+}$), ammonia ($NH_3$), and meteorological factors (temperature and RH) (Fountoukis and Nenes, 2007), and it was calculated by following equation:

$$\mathrm{pH} = -\log_{10}\left(\frac{1000 \times [H_{\mathrm{air}}^+]}{\mathrm{ALW}}\right) \quad (3)$$

where $[\text{H}_{air}^+]$ is the H$^+$ per volume of air ($\mu$g m$^{-3}$), ALW is the aerosol liquid water ($\mu$g m$^{-3}$). Here, only aerosol water associated with inorganic species is considered, as previous studies have shown that organics contribute only a small fraction (~10%) to the total ALW (Bougiatioti et al., 2016; Wang et al., 2022). $[\text{H}_{air}^+]$ and ALW can be derived directly from the model results. Due to the lack of ammonia (NH$_3$) observations during the sampling period, we used NH$_3$ (Model G2103, Picarro Inc., USA) from the summer of 2025 (13−26, July) to approximate the ammonia concentration levels at Mt. Daming. Although using data from a different year may introduce some uncertainty, this approximation is considered reasonable because the mountain is a high-altitude site (1500 m) with no significant local anthropogenic emission sources, and regional NH$_3$ mainly originates from natural releases. Under similar seasonal conditions, variations in local NH$_3$ concentrations are expected to be minor. Moreover, sensitivity analysis further supported that a small change in NH$_3$, leading to a bit pH variation (see Text S1 and Fig. S3 in the Supplement). Thus, the 2025 NH$_3$ data can be as an alteration to represent ambient NH$_3$ levels in Mt. Daming.

## 3 Results

### 3.1 Comparison of chemical composition and Fe solubility

Figure 1a shows the time series of TSP mass concentrations at Mt. Daming and in Hangzhou during the sampling period. The mean concentrations of aerosol particles reached 41 ± 17 $\mu$g m$^{-3}$ at Mt. Daming and 86 ± 28 $\mu$g m$^{-3}$ in Hangzhou, respectively. The loading of TSP in the upper mixing layer (Mt. Daming) was much lower than that at the ground-level (Hangzhou), indicating relatively clean conditions. In addition, relative humidity at Mt. Daming (88.1 ± 5.8%) was much higher than in the urban environment (70.5 ± 9.3%), confirming that the mountain site was consistently influenced by a more humid atmosphere (Table S1 in the Supplement). The pie charts in Fig. 1b show that sulfate accounted for 52% in the total measured inorganic ions at Mt. Daming, compared to 32% in Hangzhou. In comparison, nitrate accounted for 33% of the measured inorganic ions in Hangzhou, which is more than twice the fraction of nitrate (15%) in the upper mixing layer. Moreover, the average proportion of nitrate in Hangzhou slightly higher than that of sulfate in the total measured inorganic ions during the sampling period, unlike at Mt. Daming, where sulfate accounted for a significantly higher proportion than nitrate (Fig. 1b). Fig. 1c shows the mean concentrations of Fe$_T$ and Fe$_S$ at Mt. Daming were 292.3 ± 86.4 ng m$^{-3}$ and 25.6 ± 11.5 ng m$^{-3}$, respectively, which were significantly lower than those of 2094.9 ± 637.0 ng m$^{-3}$ and 69.9 ± 24.8 ng m$^{-3}$ in Hangzhou. However, the %Fe$_S$ at Mt. Daming was 8.7 ± 2.4%, 2−3 times higher than that of 3.3 ± 0.4% in Hangzhou.

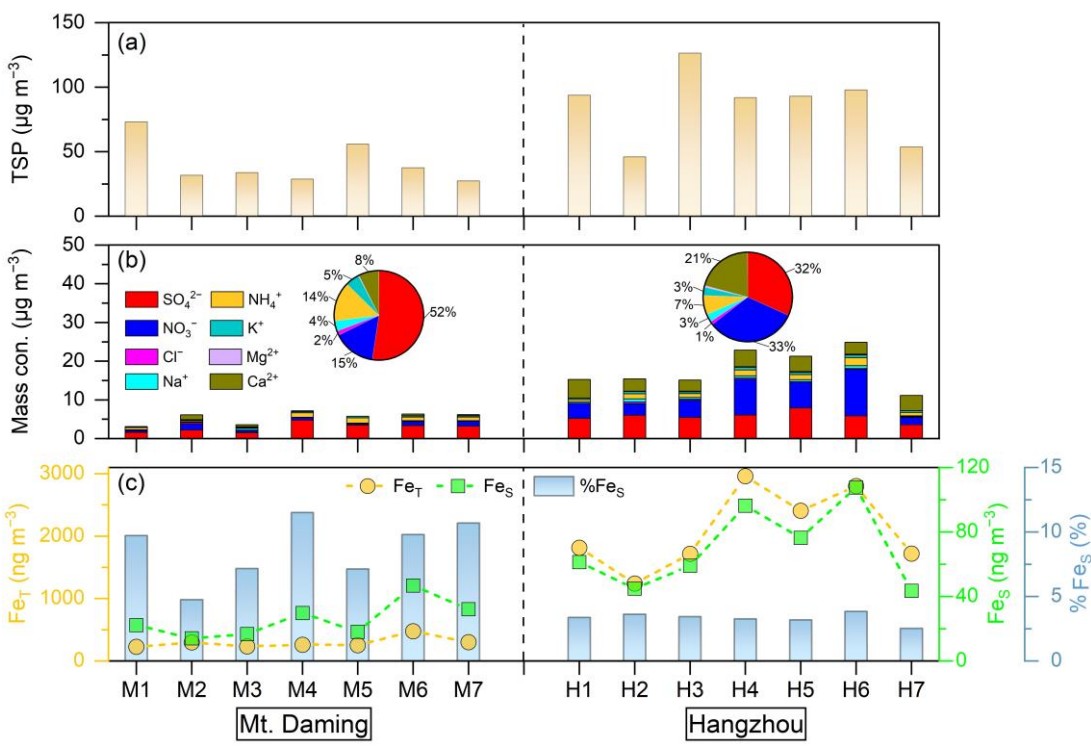

**Figure 1: Time series of (a) total suspended particle (TSP), (b) chemical species, and (c) total Fe (Fe$_T$), soluble Fe (Fe$_S$) and Fe**
**solubility (%Fe$_S$) in the upper mixing layer (Mt. Daming) and at the ground-level of Hangzhou, respectively. The labels in X-axis denote the TSP sample series at each site (M1–M7 for Mt. Daming; H1–H7 for Hangzhou).**

## 3.2 Size distributions of major chemical species and Fe

Figure 2 illustrates the size distributions of $SO_4^{2-}$, $NO_3^-$, $NH_4^+$, and $Ca^{2+}$, Fe$_T$, Fe$_S$, and %Fe$_S$ in the sized-resolved
aerosols. Since there were no MOUDI samples available in Hangzhou, we solely used MOUDI samples collected at the mountain site as an example. We observed significant differences in the size distributions of major ions. The results showed that $SO_4^{2-}$ and $NH_4^+$ both peaked at the 0.56–1.0 µm size bin (Fig. 2a–b). In contrast, $NO_3^-$ and $Ca^{2+}$ peaked at 3.2–5.6 µm and 10–18 µm size bins (Fig. 2c–d), respectively. Sulfate accounted for more than 60% of the measured inorganic ions in the submicron particles ($D_p$ <1 µm) at the mountain site, whereas nitrate contributed only 5% (Fig. S4a), consistent with the high
sulfate fraction observed in TSP at Mt. Daming (Fig. 1b). In contrast, nitrate dominated in the supermicron mode ($D_p$ >1 µm), representing 39% of the total inorganic ions and exceeding the sulfate fraction (33%) (Fig. S4b).

We further explored size distributions of Fe$_T$, Fe$_S$ and %Fe$_S$ in the upper mixing layer (Fig. 2e–f). The size distribution of Fe$_T$ was dominated by supermicron particles, with a pronounced peak in the 3.2–5.6 µm size range (Fig. 2e). The main peak aligns with the size distribution of $NO_3^-$. In contrast, the size distributions of Fe$_S$ and %Fe$_S$ exhibit similar peaks (0.56–
1.0 µm) to those of $SO_4^{2-}$ and $NH_4^+$ in the submicron fraction. The low %Fe$_S$ in supermicron particles coincided with the size

distributions of $NO_3^-$ and $Ca^{2+}$ (Fig. 2c–d). Aerosol pH was simulated using the ISORROPIA-II model. As shown in Fig. S5, supermicron particles exhibited a higher pH (4.0 ± 1.9) compared to submicron particles (2.5 ± 0.2), indicating that submicron particles were more acidic and thus more conducive to Fe acid dissolution. The implications of acid processing on aerosol Fe will be discussed later.


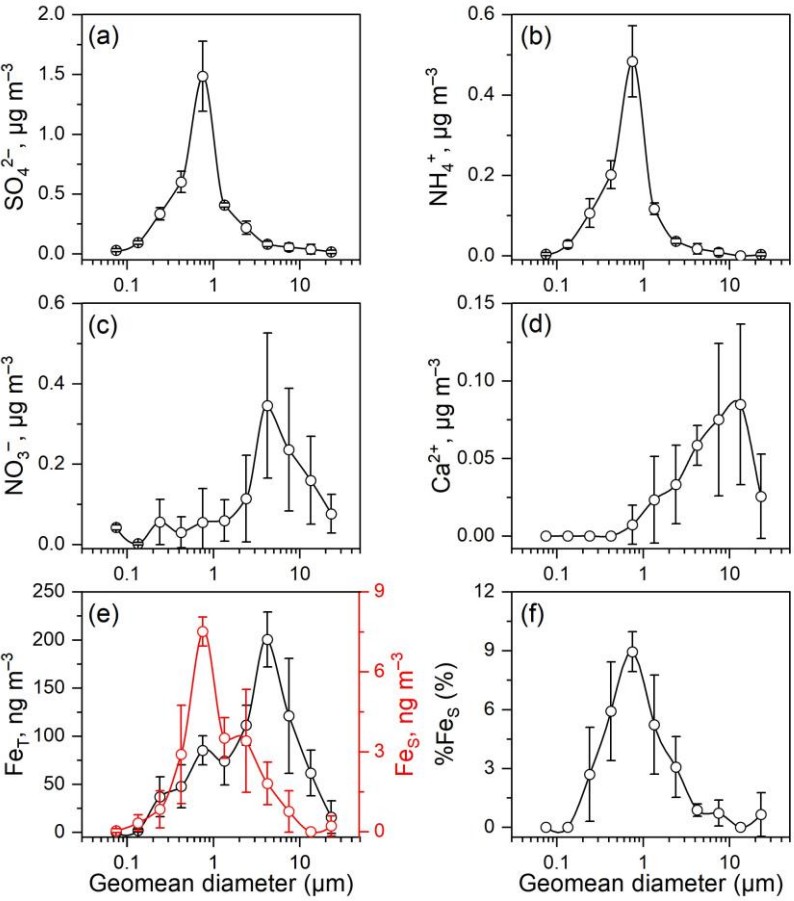

**Figure 2: Size distributions of (a) $SO_4^{2-}$, (b) $NH_4^+$, (c) $NO_3^-$, (d) $Ca^{2+}$, (e) $Fe_T$ (black) and $Fe_S$ (red), and (f) $\%Fe_S$ in the size-resolved aerosols at the upper mixing layer (Mt. Daming). The vertical bars represent one standard deviation for each diameter measurement ($n = 3$).**


### 3.3 Typical features of Fe dissolution

The ternary diagrams illustrate distinct features of $\%Fe_S$ and aerosol composition in two contrasting environments for TSP (Fig. 3) and size-resolved aerosols (Fig. 4, Mt. Daming only). As size-resolved aerosol samples were unavailable for Hangzhou, we simulated size-fractionated $\%Fe_S$ using the Integrated Massively Parallel Atmospheric Chemical Transport

(IMPACT) model (Ito and Miyakawa, 2023; Ito et al., 2019; Ito and Xu, 2014). The model description and validation were presented in Fig. S6 and Text S2 in the Supplement. Sulfate-rich particles cluster near the lower-right vertex, nitrate-rich particles near the top vertex, and ammonium- and calcium-rich particles near the left vertex of the ternary diagrams. In TSP at Mt. Daming, samples with high %$Fe_S$ but low $Fe_T$ predominantly occur near the sulfate vertex rather than the nitrate or ammonium–calcium apexes (Fig. 3), suggesting that sulfuric acid dominates Fe dissolution in the upper mixing layer. In

contrast, Hangzhou TSP samples (Fig. 3) exhibit high $Fe_T$ and low %$Fe_S$, positioned near the center or upper regions of the ternary diagram. These points are scattered toward the nitrate and ammonium–calcium apexes and are characterized by a high proportion of nitrate and alkaline species (i.e., ammonium and calcium).

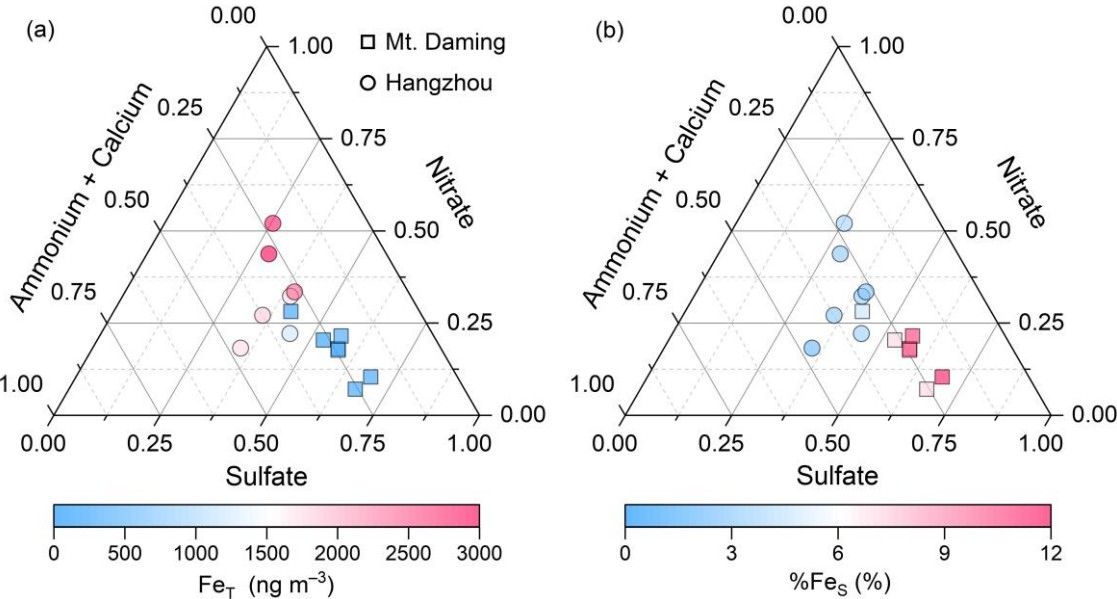

**Figure 3: The ternary diagram of the relative abundances of sulfate, nitrate, and alkaline species (ammonium + calcium) in TSP collected from the mountain site (Mt. Daming) and Hangzhou, respectively. The symbols (circles and squares) are colored by total Fe ($Fe_T$) in (a) and Fe solubility (%$Fe_S$) in (b).**

    In terms of %$Fe_S$ in size-resolved aerosols, sulfate-rich particles are primarily distributed in the submicron mode in the

upper mixing layer (Fig. 4b), whereas nitrate-rich particles mainly occur in the supermicron range. In contrast, IMPACT simulations for %$Fe_S$ in size-resolved aerosols in Hangzhou (Fig. S7) show that nitrate-rich particles are concentrated in the submicron mode, differing from that observed in the upper mixing layer. This is consistent with the higher nitrate contribution to TSP observed at the ground level in Hangzhou (Fig. 1b).

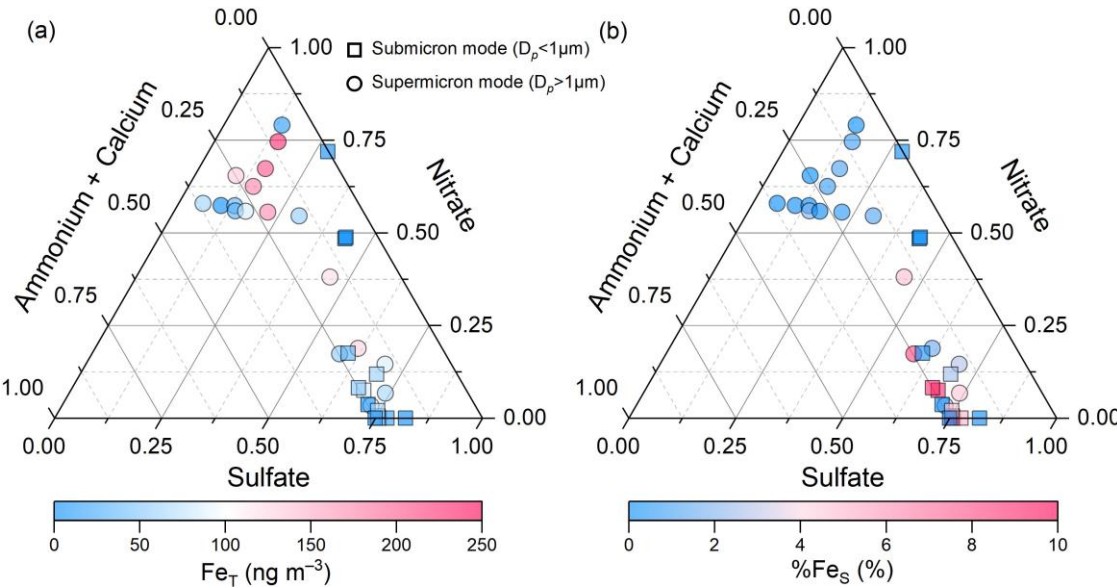

**Figure 4: The ternary diagram of the relative abundances of sulfate, nitrate, and alkaline species (ammonium + calcium) in the size-resolved aerosols collected from the upper mixing layer (Mt. Daming). The symbols (circles and squares) were colored by total Fe (Fe$_T$) in (a) and Fe solubility (%Fe$_S$) in (b).**

Figure S8 shows the correlation matrix of aerosol compositions (SO$_4^{2-}$, NO$_3^-$, NH$_4^+$ and Ca$^{2+}$), Fe$_T$, Fe$_S$, %Fe$_S$, and aerosol pH in the size-resolved aerosols collected at Mt. Daming. In the submicron particles, Fe$_T$ ($r = 0.85$, $p < 0.01$) or Fe$_S$ ($r = 0.93$, $p < 0.01$) displayed significant linear correlation with SO$_4^{2-}$, yielding positive correlation between SO$_4^{2-}$ and %Fe$_S$ ($r = 0.89$, $p < 0.01$) (left bottom panel in Fig. S8), which is consistent with previous studies (Oakes et al., 2012; Wong et al., 2020; Zhuang et al., 1992; Lei et al., 2023). Notably, no such correlations were found between NO$_3^-$ and Fe$_T$, Fe$_S$, or %Fe$_S$ in the submicron particles. This finding indicated that the sulfuric acid was the leading contributor to Fe dissolution in the submicron particles. However, correlations among aerosol compositions, Fe$_T$, Fe$_S$, %Fe$_S$, and aerosol pH in the supermicron particles showed a contrasting pattern (right panel in Fig. S8). Specifically, the correlations between SO$_4^{2-}$ and Fe$_S$ ($r = 0.81$, $p < 0.01$) or %Fe$_S$ ($r = 0.82$, $p < 0.01$) in supermicron particles were weaker than those observed in the submicron fraction. Moreover, SO$_4^{2-}$ and Fe$_T$ showed no significant correlation ($r = 0.06$, $p > 0.05$), whereas NO$_3^-$ was correlated well with Fe$_T$ but not with Fe$_S$ or %Fe$_S$. This is mainly due to the fact that total Fe is mainly derived from coarse particles such as mineral dust, as found in previous studies (Cwiertny et al., 2008; Longo et al., 2016; Chen et al., 2024b). Fe$_S$ and %Fe$_S$ in the supermicron particles exhibited significant negative correlations with aerosol pH ($p < 0.01$ and $p < 0.05$, respectively). This may suggest that the Fe dissolution is sensitive to proton levels in the supermicron particles (Zhang et al., 2022; Chen et al., 2024b). However, no such correlation was observed for submicron particles. This absence of correlation likely reflects the non-linear relationship between aerosol pH and Fe$_S$ or %Fe$_S$, as reported in previous studies (Sakata et al., 2022; Zhang et al., 2022). It is also possible that there are fewer data points to reveal the relationship between Fe dissolution and aerosol acidity.

Although a significant correlation was observed between $NH_4^+$ and %$Fe_S$, this does not necessarily indicate that $NH_4^+$ directly promotes Fe dissolution. As shown in Fig. S8, $NH_4^+$ exhibits a strong positive correlation with sulfate, but not with nitrate across both submicron and supermicron particles. This pattern likely reflects the association of ammonium with sulfate during atmospheric aging, rather than a direct pH-buffering effect. The potential role of $NH_4^+$ in Fe dissolution is further discussed in a subsequent section.

## 4 Discussion

### 4.1 Distinct aging processes of acidic species at two heights

Previous studies have shown that atmospheric aging of acidic species influences both acid types and abundance, thereby modulating Fe acidification (Hsu et al., 2010; Li et al., 2017; Srinivas et al., 2014). To delve deeper into the contribution of aging process of acidic species to Fe dissolution, the backward trajectories arriving at the upper mixing layer (Mt. Daming) and the ground-level (Hangzhou) and their corresponding molar ratios of sulfate to nitrate ($n[SO_4^{2-}]/n[NO_3^-]$) was investigated. As shown in Fig. 5a, air masses reaching Mt. Daming primarily originated from the eastern ocean and travelled through regions with relatively low $NO_x$ emissions. These trajectories, although shorter in horizontal distance, were associated with slower transport speeds and thus longer atmospheric residence times, leading to more extensive aging. Consequently, the air masses exhibited elevated $n[SO_4^{2-}]/n[NO_3^-]$ ratios (5.4 ± 3.7), mostly exceeding 3. The enhanced sulfate fraction with increasing transport time is consistent with substantial secondary sulfate formation during atmospheric aging (Longo et al., 2016; Itahashi et al., 2022; Chen et al., 2021). In contrast, air masses affecting Hangzhou mainly originated from the north and passed through nearby high-$NO_x$ emission regions (Fig. 5b). Although their horizontal pathways were longer, their faster transport speeds resulted in shorter atmospheric residence times and thus limited aging. This is reflected in the substantially lower $n[SO_4^{2-}]/n[NO_3^-]$ ratios (1.6 ± 0.7), highlighting the dominant influence of local emissions on nitrate formation.

In general, the significant contrast in $n[SO_4^{2-}]/n[NO_3^-]$ ratios between the upper mixing layer and the surface reflects distinct aging regimes. Compared to $SO_2$, $NO_x$ has a shorter atmospheric lifetime and is quickly oxidized to nitrate (Chen et al., 2021). Consequently, $n[SO_4^{2-}]/n[NO_3^-]$ typically increases with the extent of atmospheric aging. This explains the elevated ratios observed at Mt. Daming, where anthropogenic emissions are less or not anticipated. Consistent with this interpretation, $SO_2$ column mass densities at both sites were far lower than in the high-emission regions to the north and northeast (Fig. S9), indicating that local sources contribute little to sulfate levels. Backward trajectory analysis further revealed that air masses reaching both the upper mixing layer and Hangzhou had travelled regions with elevated $SO_2$ levels prior to arrival, suggesting that sulfate was predominantly formed during upwind transport. Moreover, as shown in Fig. S10, air plume heights along trajectories arriving at the upper mixing layer were generally lower than those reaching Hangzhou

and remained mostly below 1000 m, providing additional evidence that these air masses had undergone substantial atmospheric aging process within the boundary layer.

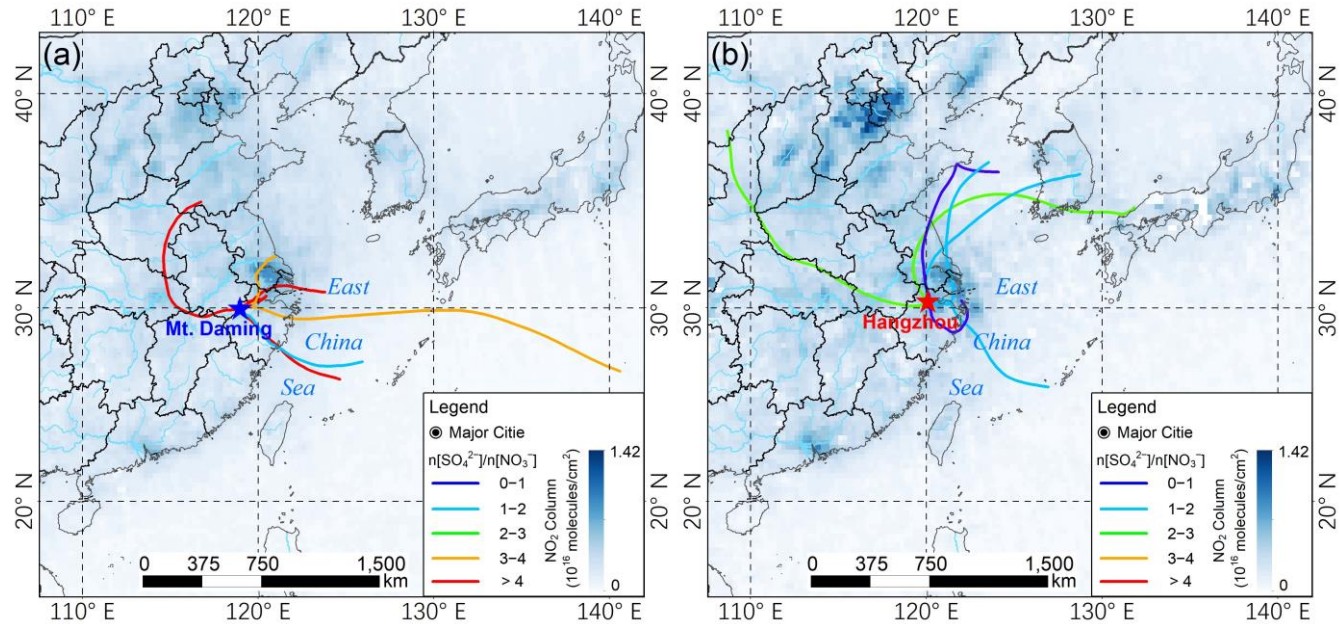

**Figure 5: 48-h backward trajectories at 500 m a.g.l. (above ground level) and their corresponding $n$[SO$_4^{2-}$]/$n$[NO$_3^-$] molar ratios during the sampling period. (a) Mt. Daming, and (b) Hangzhou. Each trajectory represents a single sample, and it is derived from the NOAA HYSPLIT Trajectory Model (available at https://www.ready.noaa.gov/HYSPLIT_traj.php, accessed on September 3, 2024). The base map shows the spatial distribution of daily averaged tropospheric NO$_2$ column concentration with a spatial resolution of 0.25°×0.25° during the sampling periods. The data was obtained from Goddard Earth Sciences Data and Information Services Center (GES DISC) (available at https://giovanni.gsfc.nasa.gov/giovanni/, accessed on October 5, 2024).**

## 4.2 Fe dissolution driven by acid processing

Field-based evidence indicates that ligand-promoted pathways involving organic acids can enhance Fe dissolution more efficiently in fine particles (Shi et al., 2022; Zhang et al., 2022). In our study, however, the analysis is based on bulk TSP samples, and oxalic acid concentrations in both the ground-level (Hangzhou) and the upper mixing layer (Mt. Daming) were relatively low (Text S3 in the Supplement). Under these conditions, Fe dissolution is likely dominated by inorganic acids, and the contribution of organic acids is therefore expected to be limited. Accordingly, we focus primarily on the proton-promoted dissolution pathway. The molar ratio of acidic species (sulfate + nitrate) to total Fe is used as a proxy to qualitatively assess the impact of aerosol acidification on %Fe$_S$ (Zhang et al., 2022; Shi et al., 2020; Liu et al., 2021; Zhu et al., 2022). Fig. 6a shows that %Fe$_S$ correlates strongly with $(n$[SO$_4^{2-}$] $+ n$[NO$_3^-$])/$n$[Fe$_T$] ($p < 0.01$), indicating that Fe dissolution is enhanced by acid processing. These ratios were higher at the mountain site than at Hangzhou, reflecting a

greater degree of Fe acidification in the upper mixing layer and explaining the vertical differences in %Fe$_S$. Prior researches indicate that %Fe$_S$ is higher under high relative humidity (RH) due to more efficient heterogeneous reactions on aqueous surfaces compared to dry particles (Shi et al., 2020; Zhu et al., 2022). Indeed, we found that the RH was normally higher (88.1 ± 5.8%) at Mt. Daming compared to RH at 70.5 ± 9.3% in Hangzhou. This suggests that the enhancement of aerosol water induced by high RH in the upper mixing layer provides an aqueous surface to foster Fe dissolution. The ISORROPIA II thermodynamic model, operated in forward mode, was used to simulate aerosol pH for TSP. The mean aerosol pH at Mt. Daming was 2.4 ± 2.3, substantially lower than that at Hangzhou (4.7 ± 2.2) (Fig. S11), indicating markedly stronger aerosol acidity in the upper mixing layer. This enhanced acidity helps explain the higher acidification potential of aerosols aloft and the correspondingly elevated %Fe$_S$ observed in this layer (Fig. 6a).

Fig. 7a presents Fe acidification process in the size-resolved aerosols. The results showed that the higher Fe acidification in submicron particles resulted in a relatively high %Fe$_S$ (3.5 ± 3.9%) in comparison to those in the supermicron particles (1.8 ± 2.2%). This disparity aligns with the size-resolved aerosol acidity measurements, which showed that submicron particles exhibited lower pH (2.5 ± 0.2) than supermicron particles (4.0 ± 1.9). Together, these observations provide additional evidence that stronger acid processing in submicron aerosols enhances Fe dissolution.

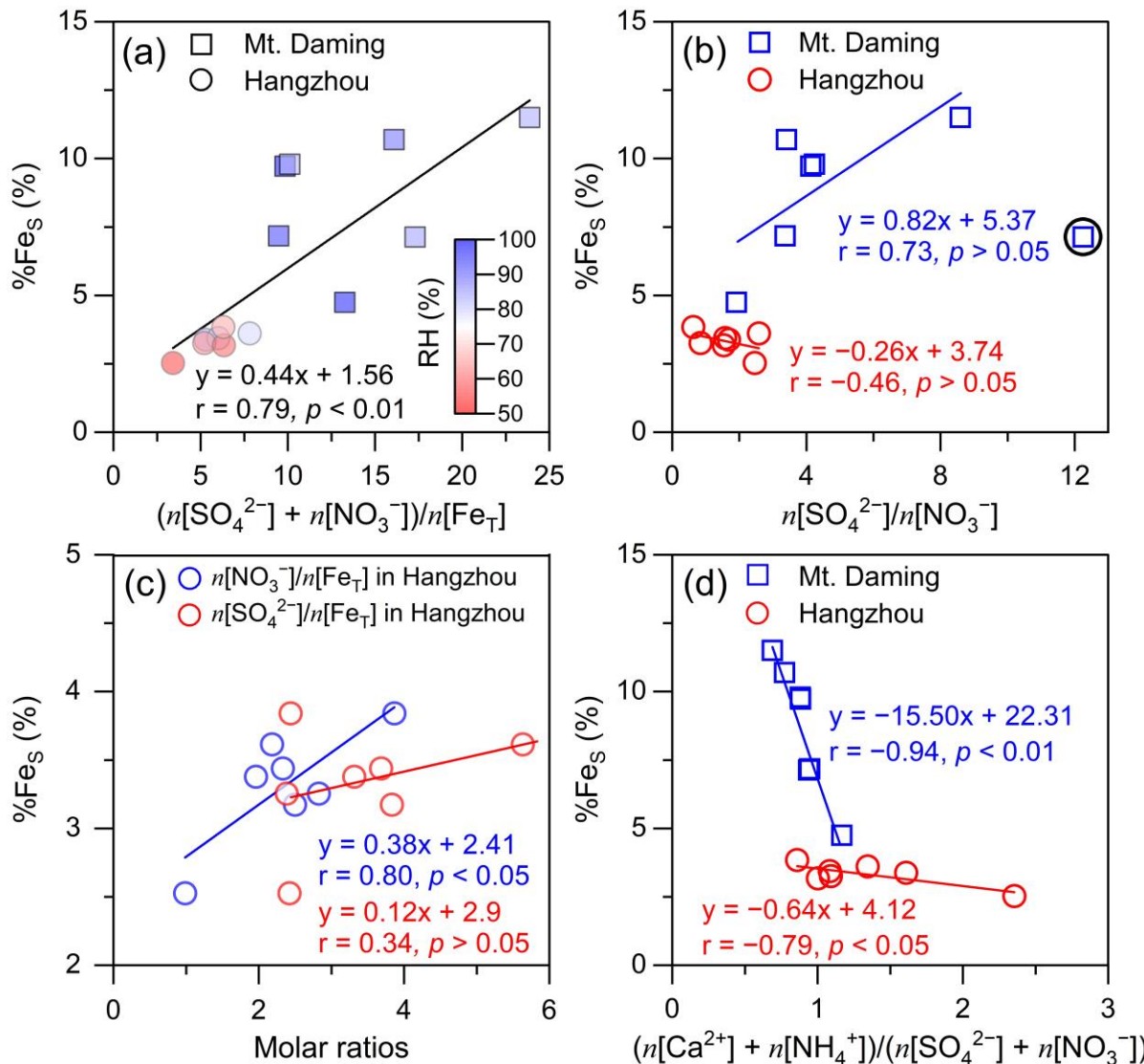

**Figure 6: Correlations between %Fe$_S$ and certain inorganic ions.** (a) $(n[SO_4^{2-}] + n[NO_3^-])/n[Fe_T]$ versus %Fe$_S$, (b) $n[SO_4^{2-}]/n[NO_3^-]$ versus %Fe$_S$, (c) $n[SO_4^{2-}]/n[Fe_T]$ or $n[NO_3^-]/n[Fe_T]$ versus %Fe$_S$ in Hangzhou, and (d) $(n[Ca^{2+}] + n[NH_4^+])/(n[SO_4^{2-}] + n[NO_3^-])$ versus %Fe$_S$ in TSP, respectively. Solid circles and squares are colored by relative humidity (RH) in (a). In plot b, one point is not included in the correlation analysis ascribed to relatively low NO$_3^-$ and is indicated by the black circle.

### 4.3 Role of sulfuric acid versus nitric acid in Fe solubility

To identify contributions of the sulfuric acid and nitric acid to %Fe$_S$, we performed an in-depth analysis of the response of Fe dissolution to acid processing driven by acidic species. The sulfate-to-nitrate molar ratio ($n[SO_4^{2-}]/n[NO_3^-]$) serves as an indicator of the relative strength and dominance of inorganic acids. Fig. 6b shows that %Fe$_S$ exhibits a positive linear

correlation with $n[SO_4^{2-}]/n[NO_3^-]$ in the upper mixing layer but no positive correlation in Hangzhou. This result suggests that sulfuric acid plays a more important role in enhancing %Fe$_S$ in the upper layer, consistent with the more prolonged atmospheric aging of air masses at higher altitudes (Section 4.1). Cwiertny et al. (2008) reported that an equivalent molar concentration of sulfuric acid dissolves ~32% more Fe from dust particles (Arizona Test Dust) than nitric acid. This might contribute to the differences we observed in our study between the sites but such an impact is not bigger enough to explain the large differences. To further reveal the contribution of these acids to Fe dissolution in Hangzhou, we examined the correlations between %Fe$_S$ and molar ratios $n[NO_3^-]/n[Fe_T]$ or $n[SO_4^{2-}]/n[Fe_T]$, following previous studies (Zhu et al., 2020; Hsu et al., 2014). our analysis revealed a significant positive correlation between $n[NO_3^-]/n[Fe_T]$ and %Fe$_S$ ($r = 0.80$, $p < 0.05$), whereas the correlation between $n[SO_4^{2-}]/n[Fe_T]$ and %Fe$_S$ was weak ($r = 0.34$, $p > 0.05$; Fig. 6c). At the mountain site, $n[NO_3^-]/n[Fe_T]$ showed no significant correlation with %Fe$_S$ ($p > 0.05$), whereas $n[SO_4^{2-}]/n[Fe_T]$ exhibited a positive, though relatively weak ($p > 0.05$), correlation with %Fe$_S$ (Fig. S12). These contrasting patterns indicate that nitric acid likely dominates Fe acidification in urban aerosols, in contrast to the sulfuric acid-driven Fe dissolution observed in the upper mixing layer.

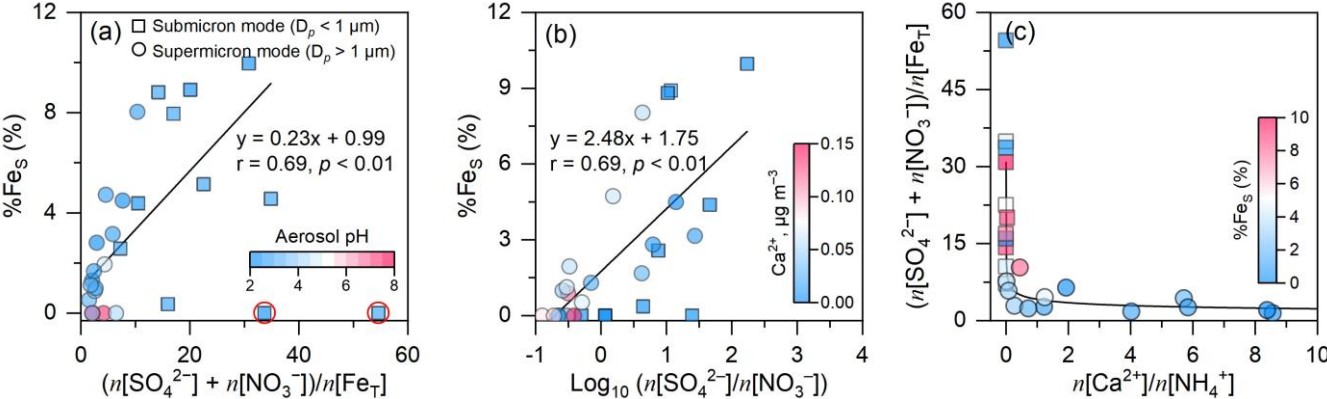

**Figure 7: Scatter plots of (a) ($n[SO_4^{2-}]$ + $n[NO_3^-]$)/$n[Fe_T]$ versus %Fe$_S$, (b) Log$_{10}$($n[SO_4^{2-}]/n[NO_3^-]$) versus %Fe$_S$, and (c) ($n[Ca^{2+}]/n[NH_4^+]$) versus ($n[SO_4^{2-}]$ + $n[NO_3^-]$)/$n[Fe_T]$), respectively, in the size-resolved aerosols at Mt. Daming. Solid circles and squares are colored by aerosol pH, Ca$^{2+}$ and %Fe$_S$ in (a), (b), and (c), respectively. In plot a, two outliers are not included in the correlation analysis due to the relatively low Fe$_S$ concentrations and are indicated by red circles.**

Similar analysis revealed a significant positive correlation ($p < 0.01$) between %Fe$_S$ and $n[SO_4^{2-}]/n[NO_3^-]$ across the size-resolved particles (Fig. 7b). When $n[SO_4^{2-}]/n[NO_3^-]$ exceeded 1 (Log$_{10}$($n[SO_4^{2-}]/n[NO_3^-]$) > 0), high sulfate concentrations in submicron particles corresponded to the elevated %Fe$_S$. In contrast, when the ratio was below 1, %Fe$_S$ in submicron particles was concentrated near the origin of the coordinate and did not exceed 3%. This comparison provides direct evidence that sulfuric acid plays a key role in enhancing Fe dissolution in the upper mixing layer, consistent with Section 4.2. Although we lacked size-resolved data for Hangzhou, urban environments in eastern China typically exhibit

high concentrations of fine-mode ammonium nitrate formed via homogeneous reactions of traffic-emitted $NO_x$ (Xie et al., 2023; Wu et al., 2020). It is expected that this fine-mode nitrate co-exists with $Fe_S$ and significantly boosts acidity (proton availability) and %$Fe_S$ in the submicron particles in urban conditions.

Here we noticed that high $Ca^{2+}$ concentrations were primarily associated with $NO_3^-$ levels and low %$Fe_S$ in supermicron particles (Fig. 7b), likely due to the buffering capacity of $Ca^{2+}$, which partially neutralizes aerosol acidity and inhibits Fe dissolution. To quantify this effect, the ratio of $(n[Ca^{2+}] + n[NH_4^+])$ to $(n[SO_4^{2-}] + n[NO_3^-])$ was used to represent as a proxy for the overall buffering capacity of alkaline species in TSP at both heights (Fig. 6d). The results showed that %$Fe_S$ was negatively correlated with $(n[Ca^{2+}] + n[NH_4^+])/(n[SO_4^{2-}] + n[NO_3^-])$ ratios at both heights, indicating that Fe dissolution is partially suppressed by the presence of alkaline species. The mean ratios were generally below 1 in the upper mixing layer $(0.9 \pm 0.1)$ but above 1 at the ground level $(1.3 \pm 0.5)$, suggesting a weaker buffering effect aloft. Further analysis of the $n[Ca^{2+}]/n[NH_4^+]$ ratio revealed the dominant alkaline species: $NH_4^+$ governed buffering in submicron particles $(n[Ca^{2+}]/n[NH_4^+]$ approaching zero), whereas $Ca^{2+}$ dominated in supermicron particles (Fig. 7c).

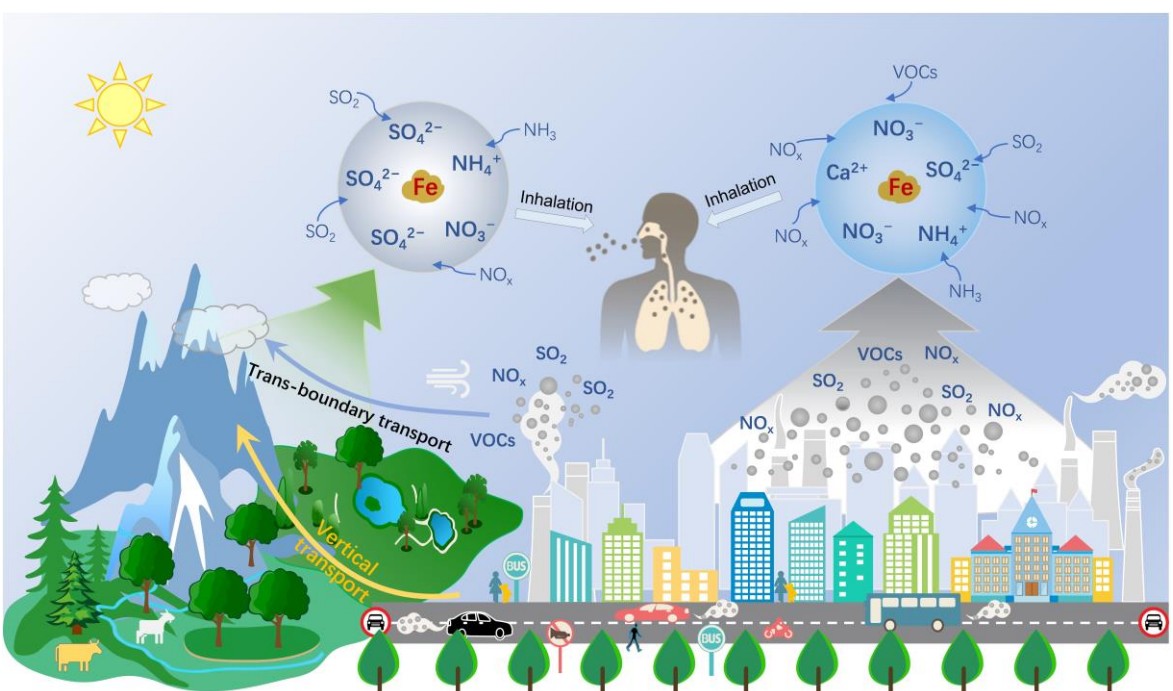

**Figure 8: Schematic illustration of atmospheric iron acid dissolution processes in the upper mixing layer and at the ground-level, and their potential implications for human health.**

## 5 Conclusions and perspectives


Our study elucidates significant vertical differences in the pathways in which Fe is dissolved by inorganic acids between the upper mixing layer and at the ground-level (Fig. 8). The dissolution of Fe depends mainly on the process of atmospheric acidification and the availability of acids in the atmosphere. The longer aging process, leading to more sulfuric vs nitric acid-dominates Fe dissolution in the upper mixing layer, in contrast to those in the urban environment. This difference leads to the

distinct %$Fe_S$ by acid processing in the two heights. Notably, nitric acid-driven Fe dissolution deserves more attention considering that $NO_x$ has replaced $SO_2$ as the dominant chemical species in most parts of China and some cities around the world (Geng et al., 2024; Van Der A et al., 2017; Ooki and Uematsu, 2005). It's projected that the contribution of the elevated nitric acid to Fe dissolution tends to become important in megacities (Ooki and Uematsu, 2005).

While the distinct contribution of sulfuric and nitric acids to Fe dissolution in the upper mixing layer and at the ground-

level is highlighted in this study, how this chemical pattern affects dissolved Fe concentration and deposition in numerical models has not been fully assessed by field observations (Liu et al., 2022; Ito, 2012). To better predict Fe dissolution and its impact on biogeochemical cycles, atmospheric chemistry models should place emphasis on the important contribution of nitric acid to Fe dissolution in the downwind locations (e.g., cities in the East Asia), where nitric acid replaces sulfuric acid as the dominant acidic species in the atmosphere (Itahashi et al., 2018; Uno et al., 2020). Although the number of samples

collected in this study is limited and does not allow assessment of long-term variability or climatological trends, our study focus on different mechanism of acid processing at the two altitudes. These field campaign provided a valuable observational dataset for testing and improving model representations of Fe dissolution. To further strength these findings, future work should incorporate long-term, vertically resolved observations to better characterize Fe solubility profiles throughout the lower troposphere.


Our focus should pivot towards the health implications stemming from Fe acidification. Previous researches have revealed that the magnetite ($Fe_3O_4$) nanoparticles produced by combustion or friction-derived heating can enter the brain directly and in turn cause damage to the human brain (Maher et al., 2016; Kirschvink et al., 1992; Lu et al., 2020). Moreover, the Fe toxicity and its valence states (Fe(II) and Fe(III)) can generate ROS in aqueous reaction, causing oxidative stress and adverse health impacts (Chen et al., 2024a; Abbaspour et al., 2014; Song et al., 2024). As far as we know, to what extent

airborne concentrations of iron nitrate affect human health is yet to be determined. Given that cities are the most densely populated and economically connected areas, traffic-related metal emissions are projected to increase. Future studies should pay more attention to the linkages between ambient nitrate, Fe dissolution, and potential adverse health impacts in urban regions (Fig. 8).

**Data availability.** The data are available upon request to the corresponding author by email.

**Author contributions.** GCW and WJL conceived the study. GCW formulated the scientific questions, performed the data analysis, and wrote the manuscript. AI performed the model simulations. CW supported the measurement of $NH_3$. XDC, BYX, CW, MKZ, KLL, LX, QY, YTW, YLS, ZBS, AI, SXZ and WJL contributed to manuscript review and editing.

**Competing interests.** The authors declare no competing interests.

**Acknowledgements.** A. I. acknowledges the Scientific Committee on Oceanic Research (SCOR) for their support of Working Group 167, Reducing Uncertainty in Soluble Aerosol Trace Element Deposition (RUSTED), via a grant to SCOR from the U.S. National Science Foundation (OCE-2513154) and the MEXT Program for the Advanced Studies of Climate Change Projection (SENTAN), Grant Number JPMXD0722681344. We also thank Yue Wang for assistance with the sampling work.

**Financial support.** This work was financially supported by the National Natural Science Foundation of China (42277080; 42561160138), National Key Research and Development Program of China (2023YFC3706301), Postdoctoral Fellowship Program of CPSF (GZC20232275), State Key Laboratory of Atmospheric Boundary Layer Physics and Atmospheric Chemistry (LAPC-KF-2023-03), Joint Funds of the Zhejiang Provincial Natural Science Foundation of China (LZJMZ24D050009).

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
