# Peer review of "Divergent iron dissolution pathways controlled by sulfuric and nitric acids from the ground-level to the upper mixing layer"

_EGUsphere, 2025_

## Author Comment (AC1)

**Response to Reviewer**

**Anonymous Referee #2**

This manuscript discusses the different dissolution pathways of Fe by nitrate and sulfate at two distinct locations. It is very interesting that sulfate dominates Fe dissolution in the upper mixing layer, while nitrate contributes to it at the ground level. The discussion integrates various insights, and I believe this paper provides important knowledge on atmospheric iron chemistry.

My most important concern is as follows. The manuscript mainly attributes the differences between the two sites to altitude. However, it may also be possible that the observed differences reflect differences in the sampling periods rather than altitude alone, and this point requires further discussion. In Figure 5, $NO_x$ emissions appear to be higher in September, when the HZ sampling was conducted than in July. Could the higher nitrate concentrations observed at HZ be influenced by such seasonal differences? I suggest that the authors demonstrate, using previous studies and/or available data, the seasonal variability of iron, TSP, nitrate, and sulfate at Damingshan and/or Hangzhou, and clarify that the observed site-to-site differences cannot be explained solely by seasonal variation.

In addition, I found it somewhat difficult to identify the fundamental factor that determines whether nitrate or sulfate contributes more strongly to iron solubility. Is it simply the difference in relative concentrations? Or could differences in their size distributions between the two sites play an important role? It would significantly strengthen the manuscript if the authors could provide a clearer discussion on what drives this difference.

**Response:** We sincerely thank the reviewer for the time and effort you have put into this review. We have carefully revised the manuscript with full consideration of the reviewer's comments and suggestions. Responses to the reviewer's comments are in ***blue***; corresponding revisions in the manuscript are in ***red*** and indented.

We thank the reviewer for raising these two critical points regarding (i) the potential influence of **seasonal variability (July vs. September)** and (ii) the **fundamental factors** controlling whether sulfate or nitrate dominates Fe dissolution in our vertical comparison.

First, as the reviewer correctly noted, synchronous observations or long-term records would be the most robust approach to fully disentangle seasonal effects from altitude-dependent processes. Unfortunately, due to logistical constraints and the unavailability of the mountain field station, we were unable to conduct additional synchronous measurements or extend the sampling period at Mt. Daming after our first campaign. Consequently, we cannot provide new in situ data for the same season at both locations.

We agree that seasonal variations may influence pollutant levels; however, we argue that the distinct Fe dissolution observed between the two sites—specifically the dominance of sulfuric acid in the upper mixing layer at Mt. Daming versus nitric acid at the ground level in Hangzhou—are ultimately governed by altitude-dependent differences in aerosol chemical composition and atmospheric aging. This interpretation is supported by the following considerations:

(1) **Vertical stratification of emission sources and atmospheric aging**

The ground-level site in Hangzhou is strongly impacted by fresh, local anthropogenic emissions, particularly traffic-related $NO_x$, which promotes rapid near-source formation of particulate nitrate. In contrast, Mt. Daming, located within the upper mixing layer, is less influenced by local emissions and instead receives regionally transported, chemically aged air masses. Under such conditions, sulfate—owing to its longer atmospheric lifetime and regional-scale formation—tends to dominate over nitrate. Our backward trajectory analysis corroborates this chemical disparity. The results showed that air masses reaching Mt. Daming were characterized by shorter pathways but longer transit times compared to the ground-level site (***Section 4.1 in Discussion***). This allowed for more extensive atmospheric aging, resulting in higher $n[SO_4^{2-}]/n[NO_3^-]$ ratios. Conversely, trajectories affecting the ground-level passed through regional $NO_x$ hotspots before reaching their destination (Hangzhou). Due to the shorter residence time, these air masses did not undergo the prolonged atmospheric aging, leading to the observed relatively lower $n[SO_4^{2-}]/n[NO_3^-]$ ratios.

Further analysis reinforces this vertically stratified aging process. In the upper mixing layer, $\%Fe_S$ increases with $n[SO_4^{2-}]/n[NO_3^-]$ and shows no significant relationship with $n[NO_3^-]/n[Fe_T]$, indicating sulfate-driven Fe dissolution. In contrast, at the ground-level, $\%Fe_S$ exhibits a strong positive correlation with $n[NO_3^-]/n[Fe_T]$, whereas no positive correlation is observed with $n[SO_4^{2-}]/n[NO_3^-]$, demonstrating the dominant role of nitric acid in promoting Fe dissolution near the surface.

Overall, these analyses indicate that although seasonal variations may affect pollutant levels, their influence on Fe dissolution is ultimately mediated through differences in aerosol chemical composition and the associated atmospheric aging processes (see ***Discussion*** section in the manuscript).

(2) **Meteorological and seasonal context**

Both July and September fall within the warm season in eastern China. Although photochemical activity and $NO_x$ emissions exhibit some seasonal variability, the fundamental vertical structure of aerosol composition remains robust: nitrate-rich aerosols near the surface and sulfate-enriched aerosols aloft due to atmospheric aging and vertical mixing. This vertical contrast has been consistently documented across different seasons and regions in eastern China, as demonstrated by paired ground-level

and mountain-based observations (e.g., Xi'an vs. Mt. Hua; Wang et al., 2013; Lake Hongze vs. Mt. Wuzhi; Zhu et al., 2016).

Taken together, we agree with the reviewer that seasonal differences may influence pollutant levels and therefore indirectly affect Fe dissolution. However, such seasonal influences act primarily through their effects on aerosol chemical characteristics and atmospheric aging process. Therefore, we argue that the observed differences in sulfate-versus nitrate-driven Fe dissolution between the two sites are best explained by altitude-dependent atmospheric aging and compositional contrasts, which form the central basis of the vertical differences highlighted in this study.

**References**

[1] Wang, G. H., Zhou, B. H., Cheng, C. L., Cao, J. J., Li, J. J., Meng, J. J., Tao, J., Zhang, R. J., and Fu, P. Q.: Impact of Gobi Desert dust on aerosol chemistry of Xi'an, inland China during spring 2009: differences in composition and size distribution between the urban ground surface and the mountain atmosphere, Atmos. Chem. Phys., 13, 819–835, https://doi.org10.5194/acp-13-819-2013, 2013.
[2] Zhu, Q., He, L.-Y., Huang, X.-F., Cao, L.-M., Gong, Z.-H., Wang, C., Zhuang, X., and Hu, M.: Atmospheric aerosol compositions and sources at two national background sites in northern and southern China, Atmos. Chem. Phys., 16, 10283–10297, https://doi.org10.5194/acp-16-10283-2016, 2016.

Second, concerning the fundamental factors controlling whether sulfate or nitrate dominates Fe dissolution, based on our analysis, the fundamental factor determining iron dissolution is the abundance of available protons ($H^+$) in atmosphere, which is governed by a combination of both the relative concentrations of acid precursors and their size distributions. We clarified this mechanism as follows:

(1) **Proton Contributions**

At the high-altitude site (Mt. Daming), the sulfate mass fraction is significantly higher than that of nitrate. Our analysis suggests that the available protons in the upper atmosphere are primarily contributed by sulfuric acid (Figure 6b-c in ***Section 4.2***). In contrast, at the urban site (Hangzhou), the nitrate fraction is nearly double that of the mountain site and slightly exceeds sulfate (Figure 1b). Here, both sulfuric and nitric acids contribute to the proton pool, but the relative contribution of nitrate becomes much more significant in the ground-level (Figure 6c).

(2) **Role of Size Distribution**.

The impact of these acids is regulated by where they reside relative to iron-containing particles: (i) At Mt. Daming (upper mixing layer): Sulfate is predominantly distributed in the submicron (fine) mode, coincident with fine Fe aerosols. Nitrate, however, is mostly found in the supermicron (coarse) mode (likely associated with mineral dust), where it is often neutralized and provides fewer free protons. Therefore, sulfuric acid

drives the dissolution in the fine mode. (ii) At Hangzhou city (ground-level): Although we lacked size-resolved data for HZ, urban environments in eastern China typically exhibit high concentrations of fine-mode ammonium nitrate formed via homogeneous reactions of traffic-emitted $NO_x$ (Wu et al., 2020; Xie et al., 2023). It is expected that this fine-mode urban nitrate co-exists with Fe and significantly boosts acidity (proton availability) in the submicron fraction.

We have revised the ***Discussion*** to explicitly address the pronounced contrast in Fe dissolution between the upper mixing layer and the surface. This revision includes a comparison of atmospheric aging and $n[SO_4^{2-}]/n[NO_3^-]$ ratios at both altitudes (***Sections 4.1 and 4.2 in Discussion***). Additionally, we added discuss how the size-resolved distribution of nitrate and sulfate potentially influences Fe dissolution at both heights (***Section 4.3 in Discussion***).

**References**

[1] Wu, C., Zhang, S., Wang, G., Lv, S., Li, D., Liu, L., Li, J., Liu, S., Du, W., Meng, J., Qiao, L., Zhou, M., Huang, C., and Wang, H.: Efficient Heterogeneous Formation of Ammonium Nitrate on the Saline Mineral Particle Surface in the Atmosphere of East Asia during Dust Storm Periods, Environ. Sci. Technol., 54, 15622–15630, https://doi.org10.1021/acs.est.0c04544, 2020.

[2] Xie, X., Hu, J., Qin, M., Guo, S., Hu, M., Ji, D., Wang, H., Lou, S., Huang, C., Liu, C., Zhang, H., Ying, Q., Liao, H., and Zhang, Y.: Evolution of atmospheric age of particles and its implications for the formation of a severe haze event in eastern China, Atmos. Chem. Phys., 23, 10563−10578, https://doi.org10.5194/acp-23-10563-2023, 2023.

**Specific Comments**

**L.68:** The manuscript states that proton concentrations for proton-promoted dissolution are estimated only by the ratios of sulfate to calcite. Is this approach universally adopted in previous studies? For example, do models such as GEOS-Chem or IMPACT use different methods?

**Response:** We appreciate the reviewer for raising this important point. The statement in the manuscript was intended to describe the simplified parameterizations only used in specific global climate models to reduce computational costs, rather than implying this is a universal approach across all atmospheric models. As mentioned in the text and correctly noted by the reviewer, schemes like the Mechanism of Intermediate Complexity for Modelling Iron (MIMI, Hamilton et al., 2019) and the Bulk Aerosol Module Iron (BAM-Fe, Scanza et al., 2018) follow that used in the Model of Atmospheric Transport and Chemistry (MATCH) by Luo et al. (2005) to set the pH values for proton-promoted dissolution rates using the ratio of sulfate to calcite. We fully agree that more sophisticated chemical transport models, such as GEOS-Chem and IMPACT (the latter of which is used in this study), utilize advanced thermodynamic equilibrium modules (e.g., ISORROPIA) to explicitly calculate proton concentrations ($H^+$) by incorporating a full suite of ions, including sulfate, nitrate, and ammonium.

This explicit approach allows for a more accurate simulation of Fe dissolution, particularly under regimes where nitric acid is a significant contributor to acidity.

We have revised the sentence in manuscript to explicitly specify that this simplified approach is limited to "certain global aerosol models that simplify the pH calculation for proton-promoted dissolution scheme," distinguishing them from chemical transport models that use explicit thermodynamic calculations. This will prevent any ambiguity regarding the methods used in models like IMPACT or GEOS-Chem.

**Pages 2−3, lines 68−71:**

"However, some global aerosol models (e.g., MATCH, MIMI and BAM-Fe) simplify the calculation of pH values for proton-promoted Fe dissolution by setting the pH solely from the sulfate-to-calcite ratio (Luo et al., 2005; Hamilton et al., 2019; Scanza et al., 2018)."

**References**

[1] Luo, C., Mahowald, N. M., Meskhidze, N., Chen, Y., Siefert, R. L., Baker, A. R., and Johansen, A. M.: Estimation of iron solubility from observations and a global aerosol model, J. Geophys. Res., 110, D23307, https://doi.org10.1029/2005JD006059, 2005.

[2] Hamilton, D. S., Scanza, R. A., Feng, Y., Guinness, J., Kok, J. F., Li, L., Liu, X., Rathod, S. D., Wan, J. S., Wu, M., and Mahowald, N. M.: Improved methodologies for Earth system modelling of atmospheric soluble iron and observation comparisons using the Mechanism of Intermediate complexity for Modelling Iron (MIMI v1.0), Geosci. Model Dev., 12, 3835−3862, https://doi.org10.5194/gmd-12-3835-2019, 2019.

[3] Scanza, R. A., Hamilton, D. S., Perez Garcia-Pando, C., Buck, C., Baker, A., and Mahowald, N. M.: Atmospheric processing of iron in mineral and combustion aerosols: development of an intermediate-complexity mechanism suitable for Earth system models, Atmos. Chem. Phys., 18, 14175−14196, https://doi.org10.5194/acp-18-14175-2018, 2018.

**L.74:** Since a main objective of this study is the comparison between the ground level and the upper mixing layer, I suggest briefly explaining why the upper mixing layer is particularly important and why it is the focus of this study.

**Response:** We thank the reviewer for this helpful suggestion. We have revised the manuscript to clarify the scientific importance of the upper mixing layer and the rationale for focusing on it.

**Page 3, lines 75−80:**

"Furthermore, existing research predominantly focuses on surface-level chemistry, neglecting the upper mixing layer where regionally aged aerosols reside. Because vertical transport significantly alters aerosol composition, the lack of altitude-resolved data limits the accuracy of atmospheric models. Investigating this vertical disparity is essential to constrain altitude-dependent mechanisms and improve model accuracy.

Here, we raise the issue of whether the key chemical processes governing Fe dissolution differ between near-surface and the upper mixing layer in eastern China."

**L.76:** The abbreviation "DMS" is used for Damingshan. In atmospheric chemistry, this could be confused with dimethyl sulfide. Consider using the site name without abbreviation.

**Response:** We thank the reviewer for this helpful suggestion. To avoid potential confusion with dimethyl sulfide (DMS) in atmospheric chemistry, we have replaced the abbreviation "DMS" with the name of Mt. Daming throughout the manuscript.

**L.90:** Please briefly describe how the mixing layer height (MLH) was determined.

**Response:** The mixing layer height (MLH) was derived from measurements obtained using a ceilometer (CL51, Vaisala, Finland). We have added this briefly description to the main text.

**Page 3, line 95−97:**

"Since mixed-layer height (MLH) data were not available at Mt. Daming during the sampling period, we referenced MLH observations (CL51, Vaisala, Finland) from Hangzhou."

**L.97:** Quartz filters may have relatively high blank levels for trace elements compared with other materials (e.g. PTFE). Why did you choose quartz filters?

**Response:** We agree that PTFE filters are often preferable for trace element analysis due to their lower blank levels. However, PTFE filters were not available during the sampling period, and quartz filters were therefore used. Although quartz filters have relatively high background values for trace elements, all sample mass concentrations were blank-corrected using field blanks to minimize potential interference from filter background levels.

**L.115:** Non-sea-salt sulfate does not appear in the results and discussion. Please check whether this description is necessary or not.

**Response:** Backward trajectory analysis indicates that air masses influencing both ground-level and mountain sites frequently passed over marine regions. To reduce marine contributions, the sea-salt fraction of sulfate was removed. Therefore, unless otherwise specified, sulfate referred to throughout the manuscript represents non-sea-salt sulfate ($nss\text{-}SO_4^{2-}$). This clarification has been explicitly stated in the text.

**Page 4, line 128:**

"Therefore, the sulfate used in the analysis is $nss\text{-}SO_4^{2-}$."

**L.140:** It would be helpful to include filter blank values normalized by filter area (ng cm$^{-2}$) in addition to ng m$^{-3}$. Please also clarify how the air volume (m$^3$) was assumed in calculating ng m$^{-3}$, and indicate the contribution of filter blank to the samples.

**Response:** Thank you for this helpful suggestion. We have revised the manuscript to report filter blank values normalized by filter area (ng cm$^{-2}$), as recommended. Soluble Fe was measured in blank filters for both TSP and size-resolved samples. In total, two sets of TSP and one set of size-resolved samples were measured as blank, yielding an average value of ~0.6 ng cm$^{-2}$. The average contribution of filter blanks to the measured samples was less than 2%. In the original manuscript, blank concentrations expressed in ng m$^{-3}$ were calculated using an average sampling air volume over the entire sampling period. We acknowledge that this approach may introduce ambiguity. To avoid this issue, we now report and apply area-normalized blank values.

**Page 5, lines 151−153:**

"… (4) the average soluble Fe on blank filters was ~0.6 ng cm$^{-2}$, determined from two sets of TSP and one set of size-resolved samples. The average contribution of filter blanks accounted for less than 2% of the measured sample concentrations."

**L.146:** the soluble Fe and total Fe 'concentration', respectively

**Response:** Revised accordingly.

**L.157:** Is the NH$_3$ data in 2025 from HZ or DMS?

**Response:** The NH$_3$ data in 2025 were obtained from the Mt. Daming. We have clarified this explicitly in the revised manuscript to avoid any ambiguity.

**Page 6, line 170:**

"…to approximate the ammonia concentration levels at Mt. Daming."

**L.164:** How was TSP measured? Please add a brief description of the method.

**Response:** Thank you for raising this question. Total suspended particles (TSP) were determined gravimetrically using an ultra-high-resolution balance (Sartorius Lab Instruments GmbH & Co, Germany). Prior to sampling, blank quartz filters were conditioned for 24 h under controlled temperature and relative humidity (25 °C and 50% RH) and then weighed. After sampling, the filters were reconditioned under the same conditions and reweighed. The difference between the post-sampling and pre-sampling filter masses was taken as the TSP mass. A brief description of TSP measurement was added to the ***Data and methods*** section.

**Page 4, lines 115−118:**

"TSP were determined gravimetrically using an ultra-high-resolution balance (Sartorius Lab Instruments GmbH & Co, Germany). Prior to sampling, blank quartz filters were conditioned for 24 h under controlled temperature and relative humidity (25 °C and 50% RH) and then weighed. After sampling, the filters were reconditioned under the same conditions and reweighed. The difference between the post-sampling and pre-sampling filter masses was taken as the TSP mass."

**Figure 2:** Does the deviation represent variability among the three sample sets? If so, I recommend adding a brief explanation.

**Response**: Yes, the deviation represents the variability among the three sample sets. We have added a brief explanation to the figure caption to clarify this point (**Page 8, line 219**).

**Figure S3:** Panels (a) and (b) are missing from the figure.

**Response**: Added accordingly.

**L.210:** In "DMS" TSP samples…

**Response**: Revised accordingly (**Page 9, line 228**).

**L.223–226:** The author discusses differences between IMPACT model results and observations at different locations. It is difficult to know if it is because of the model bias or location-dependent effects. Ideally, both sites should be compared using observations. At least, it would be helpful to demonstrate that the IMPACT model reasonably reproduces the size distributions in Figure 4, in addition to nitrate and sulfate concentrations (in TSP at HZ), before making this comparison.

**Response**: We thank the reviewer for this insightful comment. We fully agree that, ideally, both the ground-level (Hangzhou) and upper mixing layer (Mt. Daming) sites should be evaluated using size-resolved observational data. Unfortunately, due to instrument maintenance during the campaign period, size-segregated aerosol samples were not available at the Hangzhou. As a result, direct observational comparison of size-resolved Fe dissolution between the two sites was not feasible. As an alternative and following the reviewer's suggestion, we evaluated whether the IMPACT model can reasonably reproduce key aerosol chemical species before using it to support site-to-site comparison. Specifically, we first compared IMPACT-simulated sulfate and nitrate concentrations in total suspended particles (TSP) at Hangzhou with corresponding observations. The model reproduces both the magnitude and relative contributions of sulfate and nitrate reasonably well (Figure S6a–b), indicating that it captures the dominant acidifying components controlling aerosol acidity at this site.

[Figure]

**Figure S6.** Comparison of observed $SO_4^{2-}$, $NO_3^-$, and Fe solubility (%Fe$_S$) in TSP with those predicted by the Integrated Massively Parallel Atmospheric Chemical Transport (IMPACT) model over Hangzhou city. Panels show **(a)** $SO_4^{2-}$, **(b)** $NO_3^-$, and **(c)** %Fe$_S$. The solid line represents the 1:1 reference line. The dashed lines denote deviations from the solid line by a factor of $\pm$ 10.

We further compared the modeled Fe solubility with observations at Hangzhou. As shown in Figure S6c, the IMPACT model slightly overestimates Fe solubility relative to measurements; however, the discrepancy remains within one order of magnitude ($\pm 10$), which is comparable to uncertainties reported in previous global and regional Fe dissolution modeling studies. This level of agreement supports the robustness of IMPACT in simulating Fe solubility under polluted urban conditions.

Taken together, while we acknowledge the limitation associated with the lack of size-resolved observations at Hangzhou, the reasonable agreement between modeled and observed sulfate, nitrate, and Fe solubility lends confidence to the model's ability to represent key physicochemical processes relevant to Fe dissolution. We therefore believe that the IMPACT-based comparison provides meaningful insight into location-dependent differences in Fe dissolution mechanisms between the ground level and the upper mixing layer. We have incorporated this comparison (Figure S6) into the Text S2 in the ***Supplement***.

**Figure S6:** Please use either "sulfate" or "sulphate" consistently.

**Response**: Revised accordingly. We have unified the word expression of "sulfate" in the manuscript and supplement.

**L.240:** In Fig. S7, the correlations between $SO_4^{2-}$ and Fe$_S$ or %Fe$_S$ appear similarly strong for both supermicron and submicron particles. Please check this carefully. If the correlation is indeed weaker for supermicron particles, it would be helpful to show this quantitatively in the text.

**Response**: Thank you for this helpful comment. We carefully re-examined the data and confirmed that the correlations are indeed stronger in submicron particles than in supermicron particles. Specifically, in the submicron fraction, $SO_4^{2-}$ shows strong

correlations with $Fe_S$ and $\%Fe_S$, with correlation coefficients ($r$) of 0.93 and 0.89, respectively ($n = 15$, $p < 0.01$). In contrast, for supermicron particles, the corresponding correlations decrease to 0.81 and 0.82 ($n = 18$, $p < 0.01$). We have now added these quantitative results to the manuscript to clarify the size-dependent differences.

[Figure]

**Figure.** Correlations between $SO_4^{2-}$ and (a) $Fe_S$ or (c) $\%Fe_S$ in the submicron particles ($D_p < 1$ μm, left panel) and those (b and d) in the supermicron particles ($D_p > 1$ μm, right panel).

**Page 10, lines 251−254:**

"In the submicron particles, $Fe_T$ ($r = 0.85$, $p < 0.01$) and $Fe_S$ ($r = 0.93$, $p < 0.01$) displayed significant linear correlation with $SO_4^{2-}$, yielding positive correlation between $SO_4^{2-}$ and $\%Fe_S$ ($r = 0.89$, $p < 0.01$) (left bottom panel in Fig. S8), which is consistent with previous studies (Oakes et al., 2012; Wong et al., 2020; Zhuang et al., 1992; Lei et al., 2023)."

**Page 10, lines 257−258:**

"Specifically, the correlations between $SO_4^{2-}$ and $Fe_S$ ($r = 0.81$, $p < 0.01$) or $\%Fe_S$ ($r = 0.82$, $p < 0.01$) in supermicron particles were weaker than those observed in the submicron fraction."

**L.245:** Why is the negative correlation between pH and Fe$_s$ or %Fe$_s$ not observed for submicron particles? Could mechanisms other than proton-promoted dissolution be involved?

**Response**: We thank the reviewer for this insightful question. The absence of a significant negative correlation between aerosol pH and Fe$_S$ or %Fe$_S$ in submicron particles likely reflects a combination of physicochemical complexity and data limitations, as explained below.

First, although aerosol acidity (pH) is a key factor controlling Fe dissolution, its relationship with Fe$_S$ or %Fe$_S$ is not necessarily linear. Previous studies have shown that Fe solubility increases sharply only below certain acidity thresholds (Sakata et al., 2022; Zhang et al., 2022). As noted by Zhang et al. (2021), Fe solubility typically remains low at pH > 4 and increases significantly only when pH drops below a critical threshold (typically pH < 4). Such nonlinear behavior may obscure a simple linear relationship between pH and Fe$_S$ or %Fe$_S$ in submicron particles.

Second, the identification of robust correlations in the submicron fraction was also likely constrained by the limited number of data points. As shown in the Pearson correlation matrix, only 15 data points were available for submicron particles ($D_p$ <1 μm), compared with 18 for supermicron particles ($D_p$ >1 μm). This relatively small sample size may hinder the emergence of statistically robust relationships between pH and Fe$_S$ or %Fe$_S$ in the submicron fraction.

We added these explanations into manuscript.

**Pages 10–11, lines 264–266:**

"However, no such correlation was observed for submicron particles. This absence of correlation likely reflects the non-linear relationship between aerosol pH and Fe$_S$ or %Fe$_S$, as reported in previous studies (Sakata et al., 2022; Zhang et al., 2022). It is also possible that there are fewer data points to reveal the relationship between Fe dissolution and aerosol acidity."

**References**

[1] Sakata, K., Kurisu, M., Takeichi, Y., Sakaguchi, A., Tanimoto, H., Tamenori, Y., Matsuki, A., and Takahashi, Y.: Iron (Fe) speciation in size-fractionated aerosol particles in the Pacific Ocean: The role of organic complexation of Fe with humic-like substances in controlling Fe solubility, Atmos. Chem. Phys., 22, 9461−9482, https://doi.org10.5194/acp-22-9461-2022, 2022.
[2] Zhang, H., Li, R., Dong, S., Wang, F., Zhu, Y., Meng, H., Huang, C., Ren, Y., Wang, X., Hu, X., Li, T., Peng, C., Zhang, G., Xue, L., Wang, X., and Tang, M.: Abundance and Fractional Solubility of Aerosol Iron During Winter at a Coastal City in Northern China: Similarities and Contrasts Between Fine and Coarse Particles, J. Geophys. Res.: Atmos., 127, e2021JD036070, https://doi.org10.1029/2021jd036070, 2022.

**L.247:** Compared with what?

**Response**: The aim here is to compare the relationships between aerosol pH and $Fe_S$ or $\%Fe_S$ in supermicron versus submicron particles. This comparison highlights the size-dependent effects of acidity on Fe dissolution. We have addressed this point in detail in our response to the previous comment.

**Section 4.1:** Based on Figure 5, DMS does not appear to experience longer-range transport than HZ. I suggest expanding the map range so that the full trajectories are visible and comparing transport distances explicitly. In addition, both trajectories are shown at 500 m altitude; for a mountain site, trajectories at 1000 m or 1500 m may better represent air masses reaching the upper mixing layer. These may also show longer transport pathways. I also recommend reconsidering whether the differences in trajectory direction between the two sites reflect seasonal effects, site location, or altitude, by comparing trajectories under different conditions.

**Response**: We appreciate the reviewer's insightful suggestions regarding the air mass trajectory analysis. In the revised manuscript, we have expanded the geographic extent of Figure 5, which allows for clearer visualization of the full transport pathways and more direct comparison of transport distances between the two sites.

Our updated analysis shows that the airflow reaching the upper mixing layer (Mt. Daming) generally followed shorter transport pathways with slower horizontal wind speeds (for comparable trajectory transit times) (Figure 5a). This pattern suggests that these air masses experienced longer atmospheric residence times and more extensive aging prior to arrival. Moreover, these air masses passed through regions with relatively low anthropogenic influence, and some originated over the ocean before undergoing long-range transport toward the sampling site. In contrast, air masses arriving at Hangzhou exhibited longer transport distances and traversed several adjacent anthropogenic emission hotspots—particularly $NO_x$-rich regions to the northeast (Figure 5b). Given the shorter atmospheric lifetime of $NO_x$ relative to $SO_2$, nitrate forms rapidly near emission sources, whereas sulfate continues to accumulate during long-range transport. This mechanistic difference explains the elevated $n[SO_4^{2-}]/n[NO_3^-]$ ratios at Mt. Daming ($5.4 \pm 3.7$) and the lower ratios at Hangzhou ($1.6 \pm 0.7$), where fresh local emissions dominate nitric acid formation.

[Figure]

**Figure 5.** 48-h backward trajectories at 500 m a.g.l. (above ground level) and their corresponding $n[\mathrm{SO_4^{2-}}]/n[\mathrm{NO_3^-}]$ molar ratios during the sampling period. (a) Mt. Daming, and (b) Hangzhou. Each trajectory represents a single sample, and it is derived from the NOAA HYSPLIT Trajectory Model (available at https://www. ready.noaa.gov/HYSPLIT_traj.php, accessed on September 3, 2024). The base map shows the spatial distribution of daily averaged tropospheric $NO_2$ column concentration with a spatial resolution of 0.25°×0.25° during the sampling periods. The data was obtained from Goddard Earth Sciences Data and Information Services Center (GES DISC) (available at https://giovanni.gsfc.nasa.gov/giovanni/, accessed on October 5, 2024).

In the original manuscript, our purpose was to highlight that air masses reaching the upper mixing layer have undergone more extensive atmospheric aging, whereas those arriving at ground level—especially for $NO_x$—experience a shorter aging history, leading to the dominant role of local emissions in nitrate formation. In the revised manuscript, we have updated Figure 5 and expanded the discussion to compare Mt. Daming and Hangzhou in terms of transport speed, aging processes, and their influence on the observed $n[\mathrm{SO_4^{2-}}]/n[\mathrm{NO_3^-}]$ ratios.

**Pages 11−12, lines 273−300:**

"4.1 Distinct aging processes of acidic species at two heights

Previous studies have shown that atmospheric aging of acidic species influences both acid types and abundance, thereby modulating Fe acidification (Hsu et al., 2010; Li et al., 2017; Srinivas et al., 2014). To delve deeper into the contribution of aging process of acidic species to Fe dissolution, the backward trajectories arriving at the upper mixing layer (Mt. Daming) and the ground-level (Hangzhou) and their corresponding molar ratios of sulfate to nitrate ($n[\mathrm{SO_4^{2-}}]/n[\mathrm{NO_3^-}]$) was investigated. As shown in Fig. 5a, air masses reaching Mt. Daming primarily originated from the eastern ocean and travelled through regions with relatively low $NO_x$ emissions. These trajectories, although shorter in horizontal distance, were associated with slower

transport speeds and thus longer atmospheric residence times, leading to more extensive aging. Consequently, the air masses exhibited elevated $n[SO_4^{2-}]/n[NO_3^-]$ ratios (5.4 ± 3.7), mostly exceeding 3. The enhanced sulfate fraction with increasing transport time is consistent with substantial secondary sulfate formation during atmospheric aging (Longo et al., 2016; Itahashi et al., 2022; Chen et al., 2021). In contrast, air masses affecting Hangzhou mainly originated from the north and passed through nearby high-$NO_x$ emission regions (Fig. 5b). Although their horizontal pathways were longer, their faster transport speeds resulted in shorter atmospheric residence times and thus limited aging. This is reflected in the substantially lower $n[SO_4^{2-}]/n[NO_3^-]$ ratios (1.6 ± 0.7), highlighting the dominant influence of local emissions on nitrate formation.

In general, the significant contrast in $n[SO_4^{2-}]/n[NO_3^-]$ ratios between the upper mixing layer and the surface reflects distinct aging regimes. Compared to $SO_2$, $NO_x$ has a shorter atmospheric lifetime and is quickly oxidized to nitrate (Chen et al., 2021). Consequently, $n[SO_4^{2-}]/n[NO_3^-]$ typically increases with the extent of atmospheric aging. This explains the elevated ratios observed at Mt. Daming, where anthropogenic emissions are less or not anticipated. Consistent with this interpretation, $SO_2$ column mass densities at both sites were far lower than in the high-emission regions to the north and northeast (Fig. S9), indicating that local sources contribute little to sulfate levels. Backward trajectory analysis further revealed that air masses reaching both the upper mixing layer and Hangzhou had travelled regions with elevated $SO_2$ levels prior to arrival, suggesting that sulfate was predominantly formed during upwind transport. Moreover, as shown in Fig. S10, air plume heights along trajectories arriving at the upper mixing layer were generally lower than those reaching Hangzhou and remained mostly below 1000 m, providing additional evidence that these air masses had undergone substantial atmospheric aging process within the boundary layer."

Following the reviewer's suggestion, we additionally recalculated the back-trajectories for the upper mixing layer (Mt. Daming) and the ground-level (Hangzhou) using arrival heights of 1000 m and 1500 m to more accurately represent transport into the upper mixing layer. The results showed that the spatial coverage, transport directions, and pathway patterns at 1000 m and 1500 m are highly consistent with those calculated at 500 m, suggesting minimal vertical distribution difference in air mass origin during the sampling period. Therefore, the 500 m trajectories presented in the revised Figure 5 in manuscript remain representative of the airflow influencing both the upper mixing layer and the ground level.

[Figure]

**Figure.** 48-hour backward trajectories arriving at Mt. Daming and Hangzhou during the sampling period. Panels (a) and (b) show trajectories arriving at Mt. Daming and Hangzhou, respectively, at an altitude of 1000 m, while panels (c) and (d) display those arriving at 1500 m.

**L.266:** "In" general

**Response**: Revised accordingly. Please see **page 11, line 290**.

**Figure S9:** Please unify the color-bar scale for the clarity.

**Response**: Revised accordingly. Please see Figure S10 in the ***Supplement***.

[Figure]

**Figure S10.** The heights of 48-hour backward trajectories arriving at (a) the upper mixing layer (Mt. Daming) and (b) Hangzhou. Heights are given in meters (m).

**L.288 (Text S2):** I agree with the conclusion that organic acids likely do not contribute significantly to iron dissolution. However, if oxalate concentrations are estimated from sulfate and the lack of correlation between oxalate/$Fe_T$ and %$Fe_S$ is used as evidence, this would also imply no correlation between sulfate/$Fe_T$ and %$Fe_S$ (including at DMS), which seems inconsistent with the main text's argument that sulfate contributes to iron dissolution. If oxalate was not directly measured, it may be preferable not to estimate it from sulfate. I think discussion in the third paragraph ("While organic acids can …") is enough.

**Response**: We thank the reviewer for this insightful comment. We agree that estimating oxalate concentrations from sulfate and then using the lack of correlation between oxalate/$Fe_T$ and %$Fe_S$ as evidence could lead to internally inconsistent interpretations, particularly given the demonstrated role of sulfate in promoting Fe dissolution at Mt. Daming. As oxalate was not directly measured in this study, its indirect estimation introduces additional uncertainty.

Following the reviewer's suggestion, we have removed the first two paragraphs of Text S3 in the ***Supplement*** that relied on estimated oxalate concentrations. The discussion has been streamlined to retain only the third paragraph, which qualitatively addresses the potential role of organic acids in Fe dissolution without overinterpreting indirect proxies. This revision avoids contradictory conclusions while maintaining a balanced and conservative interpretation of the results.

**Figure 6b:** For DMS, should the p-value also be reported as $p > 0.05$?

**Response**: Thank you for pointing this out. We have revised Figure 6b to report the *p*-value for Mt. Daming as $p > 0.05$, following your suggestion.

**Figure 6c:** I'm also curious about the DMS results; please consider including them.

**Response**: We thank the reviewer for this helpful suggestion. We have now included the Mt. Daming results in the ***Supplement*** (Figure S12). As shown in Figure S12, the molar ratio $n[NO_3^-]/n[Fe_T]$ shows no significant correlation with %$Fe_S$, whereas $n[SO_4^{2-}]/n[Fe_T]$ exhibits a positive, though relatively weak correlation ($p > 0.05$). These findings indicate that sulfuric acid, rather than nitric acid, plays the dominant role in promoting Fe dissolution in the upper mixing layer at Mt. Daming. This analysis has been incorporated into the revised manuscript.

**Page 15, lines 356−360:**

"At the mountain site, $n[NO_3^-]/n[Fe_T]$ showed no significant correlation with %$Fe_S$ ($p > 0.05$), whereas $n[SO_4^{2-}]/n[Fe_T]$ exhibited a positive, though relatively weak ($p > 0.05$), correlation with %$Fe_S$ (Fig. S12). These contrasting patterns indicate that nitric acid likely dominates Fe acidification in urban aerosols, in contrast to the sulfuric acid-driven Fe dissolution observed in the upper mixing layer."

[Figure]

**Figure S12.** Correlations between %$Fe_S$ and $n[SO_4^{2-}]/n[Fe_T]$ or $n[NO_3^-]/n[Fe_T]$ molar ratios at Mt. Daming. Linear regression lines are shown for each relationship.

**Figure 7a:** Rather than low $Fe_T$ combined with high $(n[SO_4^{2-}] + n[NO_3^-])/n[Fe_T]$, could the issue be that $Fe_S$ was too low to allow %$Fe_S$ to be determined? Please clarify.

**Response**: We appreciate the reviewer's careful examination of Fig. 7a. We rechecked the underlying data and confirmed that the two outliers with near-zero %$Fe_S$ values (highlighted in Fig. 7a) result primarily from extremely low soluble iron ($Fe_S$) concentrations that approach the analytical detection limit. We have clarified this point

in the revised manuscript by explicitly stating that these outliers arise from low $Fe_S$ concentrations.

**Page 15, lines 365−366:**

"In plot a, two outliers are not included in the correlation analysis due to the relatively low $Fe_S$ concentrations and are indicated by red circles."

**Figure 7b:** The x-axis is shown on a logarithmic scale, but was the correlation analysis performed in linear space as in Fig. 6b? Please confirm that the correlation methods are applied consistently.

**Response**: We thank the reviewer for raising this point. In Fig. 7b, the x-axis is plotted on a logarithmic scale because the size-resolved $SO_4^{2-}/NO_3^-$ ratios span nearly four orders of magnitude (~0.1–172). Performing correlation analysis in linear space under these conditions would produce highly skewed distributions and give disproportionate weight to extreme values. Accordingly, the regression in Fig. 7b was performed using $log_{10}$-transformed $SO_4^{2-}/NO_3^-$ ratios. In contrast, Fig. 6b presents relationships within a single size range (TSP), where the variable ranges are much narrower; linear-space regression is appropriate in this case and was applied consistently. Although the data processing differs between figures, the objective in both cases is the same: to examine the relative contributions of sulfuric and nitric acid to Fe dissolution.

---

## Author Comment (AC2)

**Response to Reviewer**

**Anonymous Referee #1**

This paper investigated the mechanisms by which sulfuric and nitric acids influence iron (Fe) dissolution at different altitudes through a comparative analysis of aerosol samples collected at ground level in Hangzhou and in the upper mixing layer at Mountain Daming. It reported that there were significant vertical differences in Fe solubility and related aging processes of acidic species. The results are interesting and will help improve our understanding on the biogeochemical cycling of atmospheric Fe. Generally, this manuscript is well structured and written, I will be happy to recommend this manuscript for publication after a minor revision.

**Response**: We would like to thank the reviewer for their time and the constructive comments on our manuscript. We have carefully considered all the comments and suggestions. Below is our point-by-point response detailing how we will address each issue in the revised manuscript. In the following, paragraphs in **black** are reviewer comments; paragraphs in **blue** are point to-point responses; paragraphs in **red** are revised in the manuscript.

**Major Comments**

**Comment 1:** Line 100−105: Only 7 TSP samples per site and 3 sets of MOUDI samples (DMS only, none for HZ) are mentioned. The limitations of this sample size and its potential impact on statistical significance should be more explicitly addressed in the discussion.

**Response:** We acknowledge the reviewer's concern regarding the sample size. As noted in the original manuscript (Lines 99–101), the sampling campaign was conducted during the summer rainy season, which was frequently interrupted by rain events to prevent wet deposition contamination. Additionally, instrument maintenance prevented the collection of MOUDI samples at the Hangzhou site.

In the revised manuscript, we add a dedicated statement in the **Discussion** section to explicitly address this limitation.

**Page 17, line 404−409:**

"Although the number of samples collected in this study is limited and does not allow assessment of long-term variability or climatological trends, our study focus on different mechanism of acid processing at the two altitudes. These field campaign provided a valuable observational dataset for testing and improving model representations of Fe dissolution. To further strength these findings, future work should

incorporate long-term, vertically resolved observations to better characterize Fe solubility profiles throughout the lower troposphere."

**Comment 2:** Brief summary on air pollution conditions during the sampling period should be supplemented.

**Response:** We agree that providing a context for the air pollution levels is beneficial. We have added a brief summary of the air quality conditions in the ***Section 3.1***.

**Page 6, line 179−184:**

"Figure 1a shows the time series of TSP mass concentrations at Mt. Daming and in Hangzhou during the sampling period. The mean concentrations of aerosol particles reached $41 \pm 17 \ \mu g \ m^{-3}$ at Mt. Daming and $86 \pm 28 \ \mu g \ m^{-3}$ in Hangzhou, respectively. The loading of TSP in the upper mixing layer (Mt. Daming) was much lower than that at the ground-level (Hangzhou), indicating relatively clean conditions. In addition, relative humidity at Mt. Daming ($88.1 \pm 5.8\%$) was much higher than in the urban environment ($70.5 \pm 9.3\%$), confirming that the mountain site was consistently influenced by a more humid atmosphere (Table S1 in the Supplement).

**Comment 3:** Line 156-161: For aerosol pH calculation, only size-resolved aerosols were considered, what about the TSP samples? In addition, 2025 $NH_3$ data was used to estimate 2021 levels at DMS, while acknowledged, may introduce uncertainty. I suggest the author implement a sensitivity analysis or further discussion on the potential impact of this assumption on pH calculation.

**Response:** We thank the reviewer for these valuable comments. We answer these questions in the following:

**(1) Calculation of TSP pH**

Since concurrent ambient gas-phase $NH_3$ measurements were not available for the TSP samples during the campaign, we adopted an iteration-based method to estimate its concentration (Zhang et al., 2022; Fang et al., 2017; Sun et al., 2018). This method involves the following steps: (i) The input for ISORROPIA II is initialized as the sum of measured aerosol ammonium ($NH_4^+{}_{(p)}$]) and an estimated gas-phase component (i.e., $[TNH_4] = [NH_4^+{}_{(p)}] + [NH_{3(g)}]$). (ii) The gas-phase $NH_3$ ($NH_{3(g)}$) output from the initial run is added to the original aerosol data to update the total ammonia input for the subsequent run. (iii) This process is repeated until the variance of $NH_4^+$ output mass concentrations ($L$) fall below a 0.01 threshold. As shown in the following equation:

Input $[NH_4^+{}_{(p)} + NH_{3(g)}]_{n+1} = NH_4^+{}_{(p)} + [NH_{3(g)}]_n$  (E1)

$L = ([NH_4^+{}_{(p)}]_{n+1} - [NH_4^+{}_{(p)}]_n) / [NH_4^+{}_{(p)}]_n$  (E2)

Using this method, we performed iterations ($n > 5$), however we failed to achieve convergence. As an alternate approach, Hennigan et al., (2015) concluded that if there is no gas-phase data to constrain the thermodynamic models, the use of aerosol concentrations as input in forward-mode calculations may yield a more accurate representation of aerosol pH. Followed his suggestion, we performed ISORROPIA II using aerosol mass concentrations without gas-phase as the model inputs and set the mode with forward. The simulation indicates that the mean aerosol pH at the upper mixing layer (Mt. Daming) is $2.4 \pm 2.3$, which is significantly lower than that at ground level (Hangzhou, $4.7 \pm 2.2$) (Figure S11). This higher acidity corresponds to the greater acidification capacity and elevated Fe solubility (%Fe$_S$) observed in the upper mixing layer. We have incorporated this figure and analysis into Figure S11 in the ***Supplement*** and *Section 4.2* of the ***Discussion***, respectively.

[Figure]

**Figure S11.** Correlations between %Fe$_S$ and the corresponding molar ratio of ($n[\mathrm{SO_4^{2-}}]$ + $n[\mathrm{NO_3^-}]$)/$n[\mathrm{Fe_T}]$ at Mt. Daming and Hangzhou. Linear regression lines are shown for each site. Data points (solid circles and squares) are colored according to aerosol pH.

**Page 13, lines 327−331:**

"…The ISORROPIA II thermodynamic model, operated in forward mode, was used to simulate aerosol pH for TSP. The mean aerosol pH at Mt. Daming was $2.4 \pm 2.3$, substantially lower than that at Hangzhou ($4.7 \pm 2.2$) (Fig. S11), indicating markedly stronger aerosol acidity in the upper mixing layer. This enhanced acidity helps explain the higher acidification potential of aerosols aloft and the correspondingly elevated %Fe$_S$ observed in this layer (Fig. 6a)."

**References**

[1] Zhang, H., Li, R., Dong, S., Wang, F., Zhu, Y., Meng, H., Huang, C., Ren, Y., Wang, X., Hu, X.,

Li, T., Peng, C., Zhang, G., Xue, L., Wang, X., and Tang, M.: Abundance and Fractional Solubility of Aerosol Iron During Winter at a Coastal City in Northern China: Similarities and Contrasts Between Fine and Coarse Particles, J. Geophys. Res.: Atmos., 127, e2021JD036070, https://doi.org10.1029/2021jd036070, 2022.

[2] Fang, T., Guo, H., Zeng, L., Verma, V., Nenes, A., and Weber, R. J.: Highly Acidic Ambient Particles, Soluble Metals, and Oxidative Potential: A Link between Sulfate and Aerosol Toxicity, Environ. Sci. Technol., 51, 2611−2620, https://doi.org10.1021/acs.est.6b06151, 2017.

[3] Sun, P., Nie, W., Chi, X., Xie, Y., Huang, X., Xu, Z., Qi, X., Xu, Z., Wang, L., Wang, T., Zhang, Q., and Ding, A.: Two years of online measurement of fine particulate nitrate in the western Yangtze River Delta: influences of thermodynamics and $N_2O_5$ hydrolysis, Atmos. Chem. Phys., 18, 17177-17190, https://doi.org10.5194/acp-18-17177-2018, 2018.

**(2) Sensitivity Analysis of NH₃ Data**

To evaluate the sensitivity of pH to the assumed $NH_3$ level in size-resolved aerosols, we conducted a sensitivity analysis by varying $NH_3$ concentrations over an order of magnitude while holding all other parameters constant (Figure S3). The results showed that two times (4 ppb) change in $NH_3$ can increase pH with 0.2 unit and a ten-fold (20 ppb) change in $NH_3$ induces approximately a 1-unit change in aerosol pH—consistent with previous modeling studies (Guo et al., 2017; Weber et al., 2016). Given the relatively clean background environment at our mountain sampling site, the $NH_3$ emissions is expected to exhibit minimal variation over the short sampling period, thus, the assumed $NH_3$ concentration is likely conservative and within reasonable bounds. Therefore, while we acknowledge the potential uncertainty due to the 2025 $NH_3$ measurements, we believe the simulated aerosol pH values are sufficiently representative for investigating proton-promoted processes such as Fe dissolution.

[Figure]

**Figure S3.** Modeled aerosol pH for the Mt. Daming across different particle size bins under varying $NH_3$ concentrations (2, 4, 10, and 20 ppb), simulated using the ISORROPIA II model. The lowest $NH_3$ concentration (~2 ppb) represents the average level observed during the 2025 field campaign. The pH was calculated based on

measured inorganic composition and assumed thermodynamic equilibrium under metastable conditions. Error bars represent the standard deviation of pH values within each size bin ($n = 3$).

The sensitivity analysis of aerosol pH induced by varying $NH_3$ concentration was conducted in the Figure S3 and Texts S1 in the ***Supplement***.

**Page 6, lines 174−176:**

"Moreover, sensitivity analysis further supported that a small change in $NH_3$, leading to a bit pH variation (see Text S1 and Fig. S3 in the Supplement). Thus, the 2025 $NH_3$ data can be as an alteration to represent ambient $NH_3$ levels in Mt. Daming."

**References**

[1] Guo, H., Weber, R. J., & Nenes, A. (2017). High levels of ammonia do not raise fine particle pH sufficiently to yield nitrogen oxide-dominated sulfate production. Scientific Reports, 7(1), 12109. https://doi.org/10.1038/s41598-017-11704-0

[2] Weber, R. J., Guo, H., Russell, A. G., & Nenes, A. (2016). High aerosol acidity despite declining atmospheric sulfate concentrations over the past 15 years. Nature Geoscience, 9(4), 282−285. https://doi.org/10.1038/ngeo2665.

**Comment 4:** Figure 2e: In consideration of the deviation, single peak pattern may be more suitable for size distributions of total Fe rather than bimodal pattern.

**Response**: We agree with the reviewer's comment. Given the large standard deviations (error bars), describing it strictly as "bimodal" might overstate the separation. We have revised it in the manuscript.

**Page 7, lines 206−208:**

"The size distribution of $Fe_T$ was dominated by supermicron particles, with a pronounced peak in the 3.2–5.6 μm size range (Fig. 2e)."

**Comment 5:** Line 287–289. The concentrations of oxalate were estimated by empirical relationship with sulfate, resulting in associated uncertainty. Therefore, the result "organic acids played a limited role in Fe dissolution" should be more conservative. In addition, what about the potential synergistic effect between organic and inorganic acids on Fe dissolution?

**Response**: We thank the reviewer for this constructive comment. We acknowledge that the empirical estimation of oxalate concentrations introduces inherent uncertainties. We have revised the manuscript to adopt a more conservative tone regarding the role of organic acids and have expanded our discussion on the potential synergistic effects between organic and inorganic acids (**Page 12, lines 311–316**). However, our conclusion that organic acids play a relatively limited role in Fe dissolution should be understood in the context of the following two aspects:

**(1) Particle size dependence**

Previous studies have shown that ligand-promoted dissolution by organic acids, such as oxalate, is most effective in fine particles, where oxalate can act synergistically with protons to enhance Fe solubility. For example, Zhang et al. (2021) reported a significant correlation between Fe solubility and the oxalate-to-Fe ratio in fine particles ($r = 0.34$, $p < 0.01$), but no such relationship in coarse particles ($r = 0.04$, $p > 0.05$). Moreover, Shi et al. (2022) also showed that the contribution of organics (e.g., oxalate) to Fe solubility was more significant in the fine particles ($D_p < 1$ μm). These findings suggest that organic ligands may enhance Fe dissolution in fine aerosols but exert a much weaker influence in coarse particles. As our analysis is based on total suspended particles (TSP), which are dominated by coarse-mode, the contribution of organic-acid-driven dissolution is expected to be comparatively small.

**References**

[1] Zhang, H., Li, R., Dong, S., Wang, F., Zhu, Y., Meng, H., Huang, C., Ren, Y., Wang, X., Hu, X., Li, T., Peng, C., Zhang, G., Xue, L., Wang, X., and Tang, M.: Abundance and Fractional Solubility of Aerosol Iron During Winter at a Coastal City in Northern China: Similarities and Contrasts Between Fine and Coarse Particles, J. Geophys. Res.: Atmos., 127, e2021JD036070, https://doi.org10.1029/2021jd036070, 2022.

[2] Shi, J., Guan, Y., Gao, H., Yao, X., Wang, R., and Zhang, D.: Aerosol Iron Solubility Specification in the Global Marine Atmosphere with Machine Learning, Environ. Sci. Technol., 56, 16453−16461, https://doi.org10.1021/acs.est.2c05266, 2022.

**(2) Relative Concentration Levels**

Although organic acids can contribute to Fe mobilization, their concentrations are generally much lower than those of strong inorganic acids such as sulfuric and nitric acids. For instance, Deshmukh et al. (2023) reported oxalate-to-sulfate and oxalate-to-nitrate mass ratios of approximately 1:25 and 1:4 in fine particles ($D_p < 1$ μm), and ~1:16 and ~1:18 in coarse particles ($D_p > 1$ μm), respectively. In our original manuscript, oxalate concentrations in TSP were estimated following an empirical relationship ($[C_2O_4^{2-}] = 0.05 \times [SO_4^{2-}] − 0.273$) proposed by Yu et al. (2005), yielding oxalate-to-(sulfate + nitrate) ratios of ~6% at the mountain site and ~5% at Hangzhou. These low relative loadings suggest that even with synergistic effects, the total capacity for ligand-promoted dissolution remains small. Consistent with this interpretation, our recent work (Li et al., 2025) showed that oxalate—similar to $Ca^{2+}$—is predominantly associated with coarse-mode particles, with peak concentrations in the 3.2–5.6 μm size range, in agreement with Deshmukh et al. (2023). Given that coarse particles generally exhibit lower surface-area-to-mass ratios and slower dissolution kinetics, the effectiveness of oxalate-driven ligand-promoted dissolution in TSP is expected to be reduced.

Overall, while we recognize the uncertainties associated with the indirect estimation of oxalate and the potential for synergistic acid effects, our results suggest that, for bulk

TSP in urban and mountain environments, inorganic acids dominate Fe dissolution, with organic acids playing a secondary or modulating role rather than a primary driver. We have revised the manuscript accordingly to reflect this more conservative interpretation.

**Page 12, lines 311–316:**

"Field-based evidence indicates that ligand-promoted pathways involving organic acids can enhance Fe dissolution more efficiently in fine particles (Shi et al., 2022; Zhang et al., 2022). In our study, however, the analysis is based on bulk TSP samples, and oxalic acid concentrations in both the ground-level (Hangzhou) and the upper mixing layer (Mt. Daming) were relatively low (Text S3 in the Supplement). Under these conditions, Fe dissolution is likely dominated by inorganic acids, and the contribution of organic acids is therefore expected to be limited. Accordingly, we focus primarily on the proton-promoted dissolution pathway."

**References**

[1] Deshmukh, D. K., Kawamura, K., Kobayashi, M., & Gowda, D. (2023). Changes in the Size Distributions of Oxalic Acid and Related Polar Compounds Over Northern Japan During Spring. Journal of Geophysical Research: Atmospheres, 128(11), e2022JD038461. https://doi.org/10.1029/2022jd038461

[2] Yu, J. Z., Huang, X. F., Xu, J., & Hu, M. (2005). When Aerosol Sulfate Goes Up, So Does Oxalate-Implication for the Formation Mechanisms of Oxalate. Environmental Science & Technology, 39(1), 128–133. https://doi.org/10.1021/es049559f

[3] Li, W. J., Ito, A., Wang, G. C., et al. (2025). Aqueous-phase secondary organic aerosol formation on mineral dust, National Science Review, nwaf221, https://doi.org/10.1093/nsr/nwaf221

**Minor Comments**

**Comment 1:** The relevant work of Tang et al. (2025, AMT) and Chen et al. (2024, EST) might be considered for citation in this manuscript.

**Response:** We thank the reviewer for pointing out these relevant studies. We have now added citations of Tang et al. (2025, AMT) and Chen et al. (2024, EST) and incorporated them into our revised manuscript.

**Comment 2:** Line 185 "…or 10–18…", "or" should be replaced with "and".

**Response:** Corrected. Please refer to **Page 7, line 202**.